# Toward Faithful Retrieval-Augmented Generation with Sparse Autoencoders

**Guangzhi Xiong, Zhenghao He, Bohan Liu, Sanchit Sinha, Aidong Zhang**
Department of Computer Science
University of Virginia
{guangzhi,zhenghao,bohan,sanchit,aidong}@virginia.edu

## Abstract

Retrieval-Augmented Generation (RAG) improves the factuality of large language models (LLMs) by grounding outputs in retrieved evidence, but faithfulness failures, where generations contradict or extend beyond the provided sources, remain a critical challenge. Existing hallucination detection methods for RAG often rely either on large-scale detector training, which requires substantial annotated data, or on querying external LLM judges, which leads to high inference costs. Although some approaches attempt to leverage internal representations of LLMs for hallucination detection, their accuracy remains limited. Motivated by recent advances in mechanistic interpretability, we employ sparse autoencoders (SAEs) to disentangle internal activations, successfully identifying features that are specifically triggered during RAG hallucinations. Building on a systematic pipeline of information-based feature selection and additive feature modeling, we introduce RAGLens, a lightweight hallucination detector that accurately flags unfaithful RAG outputs using LLM internal representations. RAGLens not only achieves superior detection performance compared to existing methods, but also provides interpretable rationales for its decisions, enabling effective post-hoc mitigation of unfaithful RAG. Finally, we justify our design choices and reveal new insights into the distribution of hallucination-related signals within LLMs. The code is available at https://github.com/Teddy-XiongGZ/RAGLens.

## 1 Introduction

Retrieval-Augmented Generation (RAG) has emerged as a promising paradigm for improving the factuality of large language models (LLMs) (Lewis et al., 2020). By conditioning generation on passages retrieved from external corpora, RAG systems aim to ground model outputs in verifiable evidence. However, in practice, grounding does not eliminate unfaithfulness (Magesh et al., 2025; Gao et al., 2023). Models may still contradict the retrieved content, introduce unsupported details, or extrapolate beyond what the evidence justifies (Maynez et al., 2020; Rahman et al., 2026). These faithfulness failures, commonly referred to as hallucinations in the RAG setting, undermine user trust and limit deployment in domains where faithfulness to source information is critical (Huang et al., 2025; Zakka et al., 2024).

Various approaches have been proposed to address this challenge. One direction is to fine-tune specialized detectors to distinguish faithful from unfaithful generations (Bao et al., 2024; Tang et al., 2024a). While this method provides direct supervision, its effectiveness is often constrained by the need for large amounts of high-quality annotated training data. Another line of work employs LLMs as judges, where an auxiliary LLM is prompted to assess faithfulness given the retrieved passages and generated answers (Zheng et al., 2023; Li et al., 2024). However, these approaches struggle to detect hallucinations produced by the same model and introduce significant computational overhead when relying on large-scale external LLMs. More recently, researchers have explored the use of the LLM's internal representations, such as hidden states or attention scores, to capture hallucinations directly (Han et al., 2024; Sun et al., 2025; Zhou et al., 2025). While these methods show promise, the extraction of reliable hallucination-related signals remains challenging, and detection performance is often insufficient for practical deployment.

Meanwhile, recent advances in mechanistic interpretability have shown that sparse autoencoders (SAEs) can disentangle specific, semantically meaningful features from the hidden states of LLMs (Huben et al., 2024). By enforcing sparsity, SAEs identify features that correspond to concrete concepts, as evidenced by their consistent activation across similar cases (Bricken et al., 2023; Shu et al., 2025). This property, known as monosemanticity, provides a transparent link between internal activations and model behaviors. While recent work has explored the use of SAEs to detect signals associated with generic LLM hallucinations (Ferrando et al., 2025; Suresh et al., 2025; Abdaljalil et al., 2025; Tillman & Mossing, 2025; Xin et al., 2025), hallucinations in RAG settings pose unique challenges due to the complex interplay between retrieved evidence and generated content. It remains unclear whether SAE features can effectively capture these dynamics. In this work, we directly investigate whether SAEs can identify interpretable features that are predictive of hallucinations in RAG, enabling both accurate detection and deeper insight into failure cases.

We present RAGLens, a lightweight SAE-based detector that flags unfaithful RAG outputs by leveraging LLM internal activations through a systematic pipeline of information-based feature selection and additive feature modeling. Experimental results show that RAGLens identifies features highly relevant to RAG hallucinations and achieves superior detection performance compared to existing methods when evaluated on the same LLM. We further demonstrate the interpretability of RAGLens, enabled by its additive model structure and transparent input features, and highlight how these interpretations facilitate effective post-hoc mitigation of unfaithfulness. Finally, our analyses examine the design choices underlying RAGLens, revealing that mid-layer SAE features with high mutual information about the labels are most informative for detecting RAG hallucinations, and that generalized additive models (GAMs) are particularly well-suited for mapping SAE features to hallucination predictions. To our knowledge, RAGLens is the first approach to systematically demonstrate the effectiveness of SAE features for detecting hallucinations in RAG, and to comprehensively investigate design principles for building accurate and interpretable detectors. Here is a summary of our contributions:

- We demonstrate that SAEs capture nuanced features specifically activated during RAG hallucinations, establishing a strong foundation for detecting RAG unfaithfulness from LLM internal representations.

- Building on these SAE features, we introduce RAGLens, a lightweight hallucination detector that outperforms existing methods in detection accuracy while providing transparent and interpretable feedback to aid in hallucination mitigation.

- Through detailed analyses, we justify the key design choices in RAGLens and offer new insights into the distribution of hallucination-related signals within LLMs.

## 2 RELATED WORK

Retrieval-Augmented Generation (RAG) integrates retrieval modules with large language models (LLMs) to ground responses in external knowledge sources (Lewis et al., 2020; Guu et al., 2020). This design has improved factual accuracy in tasks such as open-domain question answering, knowledge-intensive dialogue, and domain-specific search (Shuster et al., 2021; Siriwardhana et al., 2023; Xiong et al., 2024; Oche et al., 2025; Wei et al., 2025). However, RAG systems remain vulnerable to faithfulness errors: even when relevant passages are retrieved, models may contradict evidence, invent unsupported details, or extrapolate beyond the source (Niu et al., 2024; Sun et al., 2025). These failures have been studied under terms such as hallucination, ungrounded generation, or source inconsistency, and are increasingly recognized as a central obstacle to deploying RAG in real-world applications (Zhang et al., 2025; Gao et al., 2023; Elchafei & Abu-Elkheir, 2025).

To address this challenge, a growing body of work has developed detectors to judge whether a generated response is faithful to the retrieved evidence (Manakul et al., 2023; Sriramanan et al., 2024). Early approaches focused on fine-tuning specialized detectors, which can be effective but require large amounts of high-quality training data, particularly when adapting large models (Bao et al., 2024; Tang et al., 2024a). With the rise of foundation models, researchers have begun using LLMs as evaluators, prompting an auxiliary LLM to compare generated answers against source passages and determine whether hallucination occurs (Zheng et al., 2023; Bui et al., 2024). However, this often necessitates the use of larger LLMs, leading to high computational costs, sensitivity to prompt design (Wang et al., 2024), and explanations that may be plausible but do not faithfully reflect

the underlying decision process (Turpin et al., 2023). More recent studies have explored leveraging LLM internal representations for hallucination detection, but challenges such as the polysemanticity of neurons and the opacity of hidden states have limited the extraction of high-quality features, resulting in insufficient detection performance (Elchafei & Abu-Elkheir, 2025; Sun et al., 2025).

Recent research has shown that sparse autoencoders (SAEs) can expose semantically meaningful features within the hidden representations of LLMs (Huben et al., 2024; Shu et al., 2025). By constraining activations through a sparsity-inducing bottleneck, SAEs learn dictionaries of features that often correspond to human-interpretable concepts such as syntactic roles, entities, or factual attributes (Bricken et al., 2023; Gujral et al., 2025). This capability has facilitated analysis of model internals, enhanced interpretability, and even enabled targeted control of generative behavior (Shi et al., 2025). The interpretability of SAE-derived features makes them attractive for tasks where transparency is critical, such as hallucination detection.

## 3 RAGLENS: FAITHFUL RETRIEVAL-AUGMENTED GENERATION VIA SPARSE REPRESENTATION PROBING

### 3.1 PROBLEM SETTING

Following prior work (Niu et al., 2024; Song et al., 2024; Sun et al., 2025), we use "RAG" to denote any context-conditioned generation process in which an LLM produces an answer based on both a user query/instruction and a provided context. The faithfulness detection task is to determine whether the generated answer is consistent with the given context. In such tasks, each annotated instance consists of: (1) a user query or instruction $q$; (2) a set of retrieved passages $\mathcal{C}$; (3) an answer sequence $y_{1:T}$ generated by an LLM, where $T$ is the sequence length and $y_{1:t}$ denotes the prefix up to position $t$; and (4) a binary label $\ell \in \{0, 1\}$ indicating whether the answer contains hallucination relative to $\mathcal{C}$. We assume access to a frozen LLM $\Phi$ and a corresponding SAE with encoder $\mathcal{E}$ trained on the hidden states in the $L$-th layer of $\Phi$. We denote by $\Phi_L(\cdot)$ the mapping that returns layer-$L$ hidden states. Given a generation $y_{1:T}$, we obtain

$$ h_t = \Phi_L(y_{1:t}, q, \mathcal{C}), \qquad t = 1, \ldots, T, \tag{1} $$

and transform these via the SAE encoder into sparse features

$$ z_t = \mathcal{E}(h_t), \qquad z_t \in \mathbb{R}^K, \tag{2} $$

where $K$ is the size of the dictionary and only a small number of features are active at each position.

Our goal is to examine whether the SAE features contain signals that help detect hallucinations related to RAG. Section 3.2 presents our detection method, and Section 3.3 shows how the results support explanation and mitigation. An overview is shown in Figure 1.

### 3.2 HALLUCINATION DETECTION

**Instance-level Feature Summary.** Because target labels are instance-level, we summarize token-level activations into an instance representation via channel-wise max pooling:

$$ \bar{z}_k = \max_{1 \le t \le T} z_{t,k}, \qquad k = 1, \ldots, K, \tag{3} $$

where $z_{t,k}$ is the $k$-th element of $z_t \in \mathbb{R}^K$ and collect $\bar{\mathbf{z}} = (\bar{z}_1, \ldots, \bar{z}_K) \in \mathbb{R}^K$.

**Information-based Feature Selection.** We quantify the information of each pooled feature $\bar{z}_k$ ($k = 1, \cdots, K$) about the hallucination label $\ell$ using mutual information (MI):

$$ I(\bar{z}_k; \ell) = \int_{\mathbb{R}} \sum_{\ell \in \{0,1\}} p(\bar{z}_k, \ell) \log_2 \frac{p(\bar{z}_k, \ell)}{p(\bar{z}_k)\, p(\ell)} \, d\bar{z}_k. \tag{4} $$

We rank features by MI and select the top $K'$ dimensions:

$$ \mathcal{S} = \underset{|\mathcal{S}|=K'}{\arg\max} \sum_{k \in \mathcal{S}} I(\bar{z}_k; \ell), \tag{5} $$

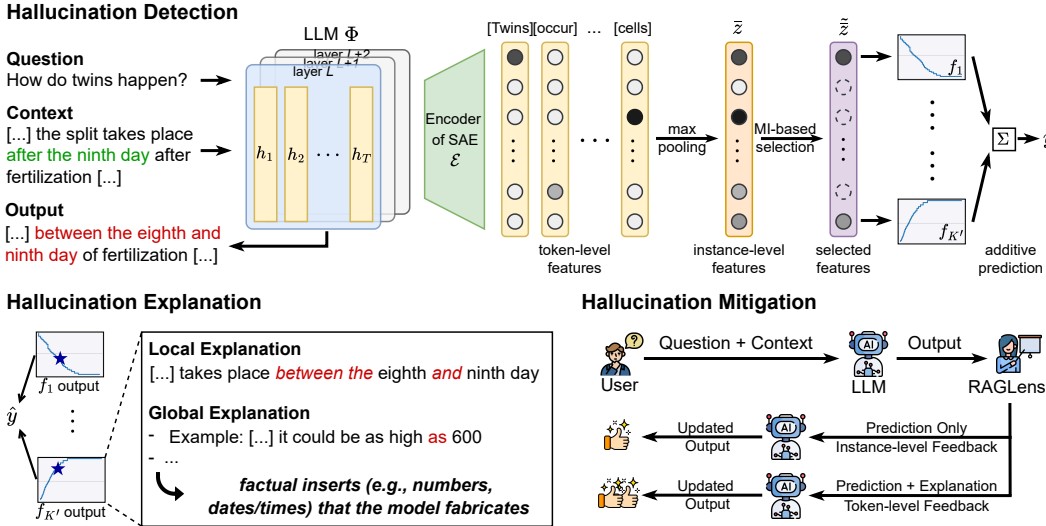

Figure 1: Overview of RAGLens for detecting, explaining, and mitigating hallucinations in retrieval-augmented generation using interpretable sparse features.

yielding $\tilde{\mathbf{z}} \in \mathbb{R}^{K'}$ as the subvector of $\bar{\mathbf{z}}$ restricted to indices $\mathcal{S}$. In our experiments, MI is estimated with a binning-based method applied to the pooled activations. While we do not explicitly utilize the hidden states of the retrieved passages $\mathcal{C}$, our encoding of the model output $y$ in Equation 1 is conditioned on $\mathcal{C}$. This allows the SAE features to implicitly capture interactions between the generated answer and the retrieved content. Empirical results in Appendix I show that the selected SAE features encode knowledge relevant to the retrieved passages, and their activations are dynamically influenced by counterfactual interventions on $\mathcal{C}$.

**Transparent Prediction with Generalized Additive Models.** After selecting informative SAE features, we model the instance label from the pooled representation using a generalized additive model (GAM) (Lou et al., 2012; Caruana et al., 2015; Hastie, 2017):

$$g(\mathbb{E}[\ell \mid \tilde{\mathbf{z}}]) \;=\; \beta_0 + \sum_{j=1}^{K'} f_j(\tilde{z}_j), \tag{6}$$

where $g$ is the link function (e.g., logit for binary classification) and each univariate shape function $f_j$ is learned using bagged gradient boosting (Nori et al., 2019). By selecting only $K'$ features ($K' \ll K$), the fitted GAM serves as a lightweight detector, requiring the encoding of only a small subset of SAE features. Our analysis in Section 5 further validates that GAM is well-suited for modeling hallucination signals from SAE features, outperforming more complex predictors such as MLP (Popescu et al., 2009) and XGBoost (Chen & Guestrin, 2016).

**Justification of Max Pooling on Sparse Activations.** Beyond the practical advantage of storage efficiency, we provide a theoretical justification for using max pooling: in the sparse activation regime, it can help distinguish hallucination-related features from random noise by amplifying signals associated with relevant targets. To facilitate the analysis, for any fixed feature index $k$, suppressing the respective notation for clarity, we model the token-level activation $z_t \geq 0$ as conditionally independent across tokens given the label $\ell \in \{0, 1\}$, with a rare activation mechanism:

$$z_t \;=\; \begin{cases} 0, & \text{with probability } 1 - p_\ell, \\ V_t, & \text{with probability } p_\ell, \end{cases} \qquad t = 1, \ldots, T, \tag{7}$$

where the "active-value" random variable $V_t$ has a distribution $F$ supported on $(0, \infty)$ that is independent of $\ell$ and i.i.d. across tokens. Let $\bar{z} = \max_{1 \leq t \leq T} z_t$ and $\pi = \Pr(\ell = 1)$.

**Theorem 1** (Max pooling in the sparse-activation regime). *If $T \times \bar{p} \ll 1$ with $\bar{p} = \frac{1}{2}(p_1 + p_0)$, then*

$$I(\bar{z}; \ell) \;=\; \frac{\pi(1 - \pi)}{2 \ln 2} \frac{T(\Delta p)^2}{\bar{p}} \;+\; O\big((T\bar{p})^2\big), \qquad \Delta p = p_1 - p_0, \tag{8}$$

*where $I(\bar{z};\ell) > 0$ iff $p_1 \neq p_0$. The leading dependence is linear in $T$ and quadratic in $\Delta p$.*

*Proof sketch.* Let $A = \mathbf{1}\{\bar{z} > 0\}$. Independence across tokens implies $\Pr(A{=}1 \mid \ell) = q_\ell = 1 - (1 - p_\ell)^T$, so $I(A;\ell) = h(\pi q_1 + (1 - \pi)q_0) - [\pi h(q_1) + (1 - \pi)h(q_0)]$. For $p_\ell \ll 1$, $q_\ell \approx T p_\ell$ and a second-order expansion of $h$ gives

$$I(A;\ell) = \frac{\pi(1 - \pi)}{2\ln 2} \frac{T\,(\Delta p)^2}{\bar{p}} + O((T\bar{p})^2). \tag{9}$$

Since $A$ is a deterministic function of $\bar{z}$, $I(A;\ell) \leq I(\bar{z};\ell)$. In the single-hit regime ($T\bar{p} \ll 1$), the extra information in $\bar{z}$ beyond $A$ occurs only on rare multi-activation events, contributing $O((T\bar{p})^2)$. Combining yields the claim. A full proof is in Appendix G. □

## 3.3 HALLUCINATION EXPLANATION AND MITIGATION

Since the detection results are computed from SAE features within an additive modeling framework, our approach naturally supports interpretability at both the local (instance-specific) and global (instance-invariant) levels, which can then be leveraged to improve faithful RAG generation.

**Local Explanations via Sparse Feature Attribution.** Because our GAM operates additively on a small set of selected SAE features, each prediction can be decomposed into a sum of feature contributions. For any given example, we can attribute the hallucination prediction to the specific sparse features that are most strongly activated. By aligning these activations with token positions, we obtain token-level feedback that highlights which parts of the generation are likely ungrounded relative to the retrieved passages. This fine-grained attribution enables users to directly pinpoint fabricated factual inserts such as numbers, dates, or named entities.

**Global Explanations via Intrinsic Model Interpretability.** Beyond instance-specific attributions, RAGLens also provides global, instance-invariant explanations. With the dictionary learning property of SAEs, each SAE feature corresponds to a semantically coherent concept that can be summarized as human-understandable knowledge. Furthermore, our use of GAMs in RAGLens enables visualization of the learned shape function for each feature, offering a stable explanation of the mapping from feature magnitude to predicted hallucination risk. Practitioners can therefore inspect how changes in a given feature systematically increase or decrease the prediction, enabling consistent feature-level auditing.

**Mitigation through Multi-level Feedback.** The interpretability of our framework enables explanation signals to be directly incorporated into mitigation strategies at inference time. Specifically, detection results can be provided to LLMs as instance-level warnings, prompting the model to reconsider and revise potentially hallucinated content. By aligning sparse activations with text spans identified by local explanations, we can further highlight problematic tokens that may require editing, thereby guiding the model to refine its output.

## 4 EXPERIMENTS

### 4.1 EXPERIMENTAL SETTINGS

We conduct experiments on two RAG hallucination benchmarks with Llama2 backbones: RAGTruth (Niu et al., 2024) and Dolly (Accurate Context) (Hu et al., 2024), both of which include human annotations for outputs generated by Llama2-7B/13B. To further evaluate generalizability across architectures, we also test our method on Llama3.2-1B, Llama3.1-8B, and Qwen3-0.6B/4B using two additional datasets, AggreFact (Tang et al., 2023) and TofuEval (Tang et al., 2024b), which contain hallucinations produced by a variety of LLMs. For consistency with prior work (Sun et al., 2025; Tamber et al., 2025), we report balanced accuracy (Acc) and macro $F_1$ ($F_1$).

We compare RAGLens with representative detectors based on (i) prompt engineering (e.g., Friel & Sanyal, 2023), (ii) model uncertainty (e.g., Manakul et al., 2023), or (iii) LLM internal representations (e.g., Sun et al., 2025). We also include a fine-tuning baseline, "Llama2-13B(LR)", following existing work (Sun et al., 2025). For a fair comparison with methods that analyze internal signals during LLM generation, we evaluate all detectors on Llama2-7B and Llama2-13B using the corresponding samples generated by these models in RAGTruth and Dolly. Table 1 lists the specific baselines, with details in Appendix A. Implementation details are provided in Appendix B.

## 4.2 Performance on RAG Hallucination Detection

Table 1 summarizes the performance of RAGLens on RAGTruth and Dolly, with comparisons to previous methods using the same backbones. As the table shows, the SAE features of both Llama2-7B and Llama2-13B contain sufficient information to accurately detect hallucinations, achieving AUC scores greater than 80% on both datasets. More importantly, RAGLens consistently outperforms existing baselines, demonstrating its effectiveness in identifying and leveraging internal knowledge for RAG hallucination detection. These results highlight the strong potential of SAEs to serve as powerful detectors by using knowledge already embedded within LLMs to identify hallucinations.

Table 1: Performance comparison of different hallucination detection methods on RAGTruth and Dolly. The best results are highlighted in **bold**.

| Method | RAGTruth (Llama2-7B) | | | Dolly (Llama2-7B) | | | RAGTruth (Llama2-13B) | | | Dolly (Llama2-13B) | | |
|---|---|---|---|---|---|---|---|---|---|---|---|---|
| | AUC | Acc | $F_1$ | AUC | Acc | $F_1$ | AUC | Acc | $F_1$ | AUC | Acc | $F_1$ |
| Prompt | – | 0.6700 | 0.6720 | – | 0.6200 | 0.5476 | – | 0.7300 | 0.6899 | – | 0.6700 | 0.5823 |
| Llama2-13B(LR) | – | 0.6350 | 0.6572 | – | 0.6043 | 0.6616 | – | 0.7044 | 0.6725 | – | 0.5545 | 0.6664 |
| LwMLM | – | 0.6940 | 0.7365 | – | 0.6550 | 0.7702 | – | 0.5956 | 0.7684 | – | 0.6800 | 0.7000 |
| FAcTScore | 0.5428 | 0.5333 | 0.6719 | 0.4813 | 0.5354 | 0.6849 | 0.5294 | 0.4533 | 0.6239 | 0.4389 | 0.4646 | 0.5954 |
| FactCC | 0.4976 | 0.5022 | 0.4589 | 0.6169 | 0.5758 | 0.5882 | 0.4753 | 0.4800 | 0.4121 | 0.6496 | 0.6162 | 0.5250 |
| ChainPoll | 0.6738 | 0.6841 | 0.7006 | 0.6593 | 0.6200 | 0.5581 | 0.7414 | 0.7378 | 0.7370 | 0.7070 | 0.6800 | 0.6004 |
| RAGAS | 0.7290 | 0.6822 | 0.6667 | 0.6648 | 0.6560 | 0.6392 | 0.7541 | 0.7080 | 0.6987 | 0.6412 | 0.6480 | 0.5306 |
| TurLens | 0.6510 | 0.6821 | 0.6658 | 0.6264 | 0.6800 | 0.6567 | 0.7073 | 0.6756 | 0.7063 | 0.6622 | 0.5700 | 0.3944 |
| RefCheck | 0.6912 | 0.6467 | 0.6736 | 0.6494 | 0.6100 | 0.5412 | 0.7897 | 0.7200 | 0.7823 | 0.6621 | 0.5700 | 0.3944 |
| P(True) | 0.7093 | 0.5648 | 0.6549 | 0.6191 | 0.5344 | 0.5095 | 0.8496 | 0.6266 | 0.7038 | 0.6422 | 0.5260 | 0.5240 |
| SelfCheckGPT | – | 0.5844 | 0.4642 | – | 0.5300 | 0.3188 | – | 0.5844 | 0.4642 | – | 0.5300 | 0.3188 |
| LN-Entropy | 0.5912 | 0.5620 | 0.6850 | 0.6074 | 0.5656 | 0.6261 | 0.5912 | 0.5620 | 0.6850 | 0.6074 | 0.5656 | 0.6261 |
| Energy | 0.5619 | 0.5088 | 0.6657 | 0.6074 | 0.5656 | 0.6261 | 0.5619 | 0.5088 | 0.6657 | 0.6074 | 0.5656 | 0.6261 |
| Focus | 0.6233 | 0.5533 | 0.6522 | 0.6783 | 0.6212 | 0.6545 | 0.7888 | 0.6000 | 0.6758 | 0.7067 | 0.6500 | 0.6567 |
| Perplexity | 0.5091 | 0.5333 | 0.6749 | 0.6825 | 0.6363 | 0.7097 | 0.5091 | 0.5333 | 0.6749 | 0.6825 | 0.6363 | 0.7097 |
| EigenScore | 0.6045 | 0.5422 | 0.6682 | 0.6786 | 0.6596 | 0.7241 | 0.6640 | 0.5267 | 0.6637 | 0.7214 | 0.6211 | 0.7200 |
| SEP | 0.7143 | 0.6187 | 0.7048 | 0.6067 | 0.6060 | 0.7023 | 0.8098 | 0.7288 | 0.7799 | 0.7093 | 0.6800 | 0.6923 |
| SAPLMA | 0.7107 | 0.5155 | 0.6502 | 0.6500 | 0.6084 | 0.6653 | 0.8029 | 0.5488 | 0.6923 | 0.7088 | 0.6100 | 0.6605 |
| ITI | 0.6714 | 0.5667 | 0.6496 | 0.5494 | 0.5800 | 0.6281 | 0.8501 | 0.6177 | 0.6850 | 0.6530 | 0.5583 | 0.6712 |
| ReDeEP | 0.7458 | 0.6822 | 0.7190 | 0.7949 | 0.7373 | 0.7833 | 0.8244 | 0.7889 | 0.7587 | 0.8420 | 0.7070 | 0.7603 |
| **RAGLens (Ours)** | **0.8413** | **0.7576** | **0.7636** | **0.8764** | **0.7778** | **0.8070** | **0.8964** | **0.8333** | **0.8148** | **0.8568** | **0.7576** | **0.7895** |

## 4.3 Cross-model Application

While SAE features are not transferable across different LLMs, the RAGLens detector trained on one LLM can be flexibly applied to text outputs generated by other LLMs. To examine whether LLMs contain sufficient internal knowledge to detect hallucinations produced by other LLMs, we conduct cross-model evaluations by training a series of RAGLens detectors based on SAEs of multiple open-source LLMs, and test their performance on RAG outputs from various LLMs in RAGTruth, AggreFact, and TofuEval. Specifically, we prompt each LLM to assess the faithfulness of the RAG output in a chain-of-thought (CoT) style (Wei et al., 2022), using the template from Luo et al. (2023), and compare these results to those of the same model's SAE-based detector via RAGLens.

Figure 2 shows the performance of all evaluated LLMs across different datasets. The SAE-based detector consistently outperforms each model's own CoT-style self-judgments. Larger LLMs exhibit stronger internal knowledge, achieving higher detection performance with SAE-based detectors. While earlier generation models such as Llama2-7B and Llama2-13B have lower CoT judgments on certain datasets, their SAE-based detectors perform comparably to newer models of similar size (e.g., Llama3.1-8B). Meanwhile, although Qwen3-0.6B achieves competitive CoT performance on AggreFact and TofuEval, its SAE-based detector lags behind those of larger LLMs, suggesting that the informativeness of internal knowledge correlates more with model size than with training pipeline. Overall, these results indicate that models "know more than they tell" and that SAEs can reveal latent faithfulness signals that are not consistently captured by CoT reasoning.

## 4.4 Generalization across Domains

Beyond cross-model applications, we further assess whether the internal signals captured by RAGLens generalize across domains. Specifically, we train the RAGLens predictor on one domain and evaluate its performance on other domains. Table 2 reports the generalization performance (AUC) of RAGLens across different datasets and tasks.

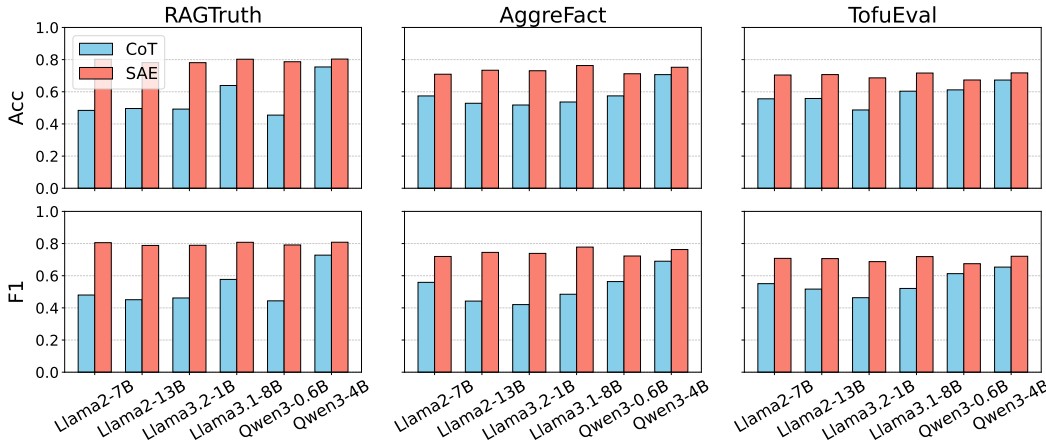

Figure 2: Comparison of LLM CoT-style self-judgment versus internal knowledge revealed by SAE features for hallucination detection across datasets.

The left part of Table 2 presents cross-dataset results, showing that RAGLens generalizability depends on the diversity of its training data. For example, a detector trained on RAGTruth significantly outperforms the CoT baseline on AggreFact and TofuEval without retraining. In contrast, predictors trained on AggreFact and TofuEval, while still outperforming CoT in most cases, do not generalize as well as those trained on RAGTruth. This can be attributed to dataset differences: RAGTruth covers multiple subtasks, whereas AggreFact and TofuEval focus on single tasks. The results indicate that RAGLens trained with diverse samples is more robust and generalizable to domain shifts.

Table 2: Generalization across datasets and subtasks. Left: RAGLens trained on a single dataset and evaluated on others. Right: RAGLens trained on one RAGTruth subtask and evaluated on other subtasks. "None" indicates zero-shot performance with CoT prompting. All scores are AUROC.

| Train\Test | RAGTruth | AggreFact | TofuEval | | Train\Test | Summary | QA | Data2txt |
|---|---|---|---|---|---|---|---|---|
| Llama2-7B | | | | | Llama2-7B | | | |
| None | 0.4842 | 0.5741 | 0.5562 | | None | 0.4924 | 0.4845 | 0.4949 |
| RAGTruth | 0.8806 | 0.8019 | 0.7637 | | Summary | 0.8191 | 0.8253 | 0.6443 |
| AggreFact | 0.5330 | 0.8330 | 0.6123 | | QA | 0.7081 | 0.8835 | 0.6609 |
| TofuEval | 0.7747 | 0.6161 | 0.7846 | | Data2txt | 0.5386 | 0.6616 | 0.8454 |
| Llama2-13B | | | | | Llama2-13B | | | |
| None | 0.4959 | 0.5285 | 0.5583 | | None | 0.5196 | 0.5088 | 0.4765 |
| RAGTruth | 0.8674 | 0.7831 | 0.7319 | | Summary | 0.7539 | 0.8330 | 0.6627 |
| AggreFact | 0.4669 | 0.8285 | 0.6239 | | QA | 0.6619 | 0.8769 | 0.6669 |
| TofuEval | 0.7342 | 0.5727 | 0.7883 | | Data2txt | 0.5653 | 0.7373 | 0.8491 |

Generalization across task types is shown in the right part of Table 2. RAGLens, when trained on one task, consistently transfers its learned knowledge to other tasks and outperforms the CoT baseline. Among the three subtasks, the predictor trained on summarization (Summary) exhibits the strongest generalizability, surpassing those trained on question answering (QA) or data-to-text generation (Data2txt). Additionally, knowledge transfer between Summary and QA is more effective than between Data2txt and the other tasks. These results suggest that RAGLens can capture common signals shared across different RAG tasks, while also revealing the presence of task-specific signals that limit generalization.

## 4.5 INTERPRETABILITY OF RAGLENS

In addition to strong performance, RAGLens provides interpretability for the hallucination detection process. Since RAGLens uses SAE features that are disentangled and correspond to specific concepts, we can analyze which features are most indicative of hallucinations and what they represent.

With the deployment of the GAM classifier, RAGLens can transparently illustrate how each feature contributes to the final prediction through learned shape functions. Table 3 presents two representative SAE features from Llama3.1-8B that are most predictive of hallucinations, as identified by the GAM classifier trained on RAGTruth. For each feature, we show two example activations from RAGTruth outputs, accompanied by a semantic explanation distilled by GPT-5 from 24 activation cases. We also visualize the learned shape function for each feature, where the y-axis (feature effect on hallucination prediction) is zero-centered, illustrating how the feature value (x-axis) influences the likelihood of hallucination.

Table 3: Interpretation of two SAE features in Llama3.1-8B. Each row shows the feature ID, a brief explanation of its semantic role, and example text spans where the feature is activated. Top activated tokens in each example are shown in **bold**, while the hallucinated tokens are highlighted in red. The shape functions learned by the GAM are visualized in the rightmost column, illustrating each feature's impact on hallucination prediction.

| ID | Explanation | Examples | Shape Plot |
|---|---|---|---|
| 22790 | unsupported numeric/time specifics | Context: no mention of age
Output: [...] at the age of 34 [...] |  |
| | | Context: no mention of release schedule
Output: [...] to be **released in** August [...] | |
| 17721 | grounded, high-salience tokens | Context: [...] could be arrested on the spot [...]
Output: [...] could be **arrested** on the spot [...] |  |
| | | Context: [...] software can be licensed as a [...]
Output: [...] software can be **licensed** as a [...] | |

As shown in the table, Llama3.1-8B contains various types of features that help detect hallucinations from different perspectives. For example, feature 22790 indicates potential hallucinations that are related to unsupported numeric/time specifics. Its corresponding shape function (learned by GAM) exhibits a monotonic increase in hallucination likelihood as activation strength rises. RAGLens also uncovers SAE features that are negatively correlated with hallucinations, such as feature 17721, which captures signals associated with grounded, high-salience tokens. This interpretability not only clarifies how RAGLens works, but also provides insights into the internal knowledge of LLMs. Additional examples from other LLMs are provided in Appendix H, and Appendix I presents case studies using counterfactual perturbations to validate that these features are specifically sensitive to hallucination patterns unique to RAG scenarios.

## 4.6 MITIGATION OF UNFAITHFULNESS WITH RAGLENS

Leveraging its detection and interpretation capabilities, RAGLens can provide post-hoc feedback to LLMs to mitigate hallucinations. We evaluate this by applying Llama2-7B-based RAGLens to 450 Llama2-7B-generated outputs from RAGTruth, and prompting the same model to revise its original output using RAGLens feedback. Specifically, we compare the effectiveness of instance-level feedback (detection results only) and token-level feedback (which includes additional explanations from RAGLens interpretation) for hallucination mitigation.

Table 4 reports the resulting hallucination rates (lower is better) as judged by multiple automatic LLM judges. In addition, two human annotators evaluated a subset of 45 outputs, with an inter-annotator agreement of 78.3%. Although hallucination rates vary among different types of annotators, the results consistently show that both types of RAGLens feedback effectively reduce hallucinations in the revised output. Notably, the more nuanced token-level feedback enabled by RAGLens interpretability leads to further reductions compared to instance-level feedback. We further applied a trained RAGLens detector (Llama3.1-8B based) to all 450 examples and found that instance-level feedback converted 29 outputs from hallucination to non-hallucination, while token-level feedback achieved 36 such conversions, confirming the advantage of token-level feedback.

Table 4: Mitigation of Llama2-7B hallucinations using SAE-based internal knowledge. Hallucination rates (lower is better) are reported for original outputs and after applying instance- and token-level feedback, as judged by Llama3.3-70B, GPT-4o, GPT-o3, and human annotators.

|                          | Llama3.3-70B | GPT-4o  | GPT-o3  | Human   |
|--------------------------|--------------|---------|---------|---------|
| Original                 | 43.78%       | 37.78%  | 64.44%  | 71.11%  |
| + Instance-level Feedback| 42.22%       | 36.44%  | 60.44%  | 62.22%  |
| + Token-level Feedback   | **39.11%**   | **34.22%** | **58.88%** | **55.56%** |

## 5  DISCUSSIONS

Beyond the main results on hallucination detection, interpretation, and mitigation using SAE features, we further analyze several key SAE-specific design choices in RAGLens, including the selection of the LLM layer, the feature extractor, the number of selected features, and the predictor architecture.

**LLM Layer Selection.** We first vary the layer from which SAE features are extracted, covering the full depth of several LLMs (Llama3.2-1B, Llama3-8B, Qwen3-0.6B, and Qwen3-4B). Figure 3 presents the heatmaps of LLM performance on various subtasks in RAGTruth (RAGTruth-Summary, RAGTruth-QA, and RAGTruth-Data2txt), where layer depths are normalized for direct comparison. The results show that the performance trend in layers is consistent among LLMs but varies by task. In the Summary and QA tasks of RAGTruth, the performance peaks around the middle layers, whereas the Data2txt task exhibits a comparatively flat performance pattern across layers.

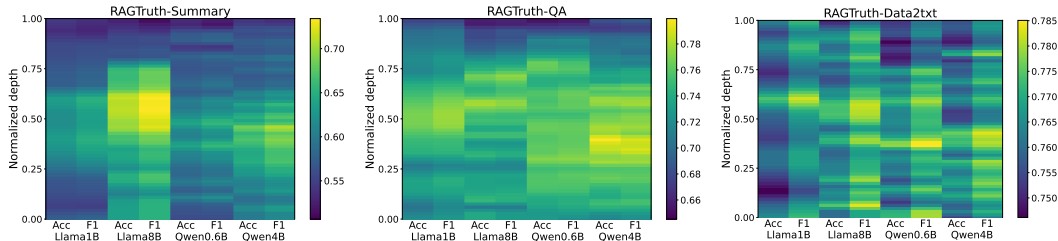

Figure 3: Layer-wise analysis of Llama3.2-1B, Llama3-8B, Qwen3-0.6B, and Qwen3-4B on subtasks in RAGTruth (RAGTruth-Summary, RAGTruth-QA, and RAGTruth-Data2txt).

**Feature Extractor Comparison.** We further compare different feature extractors, specifically SAE and Transcoder (Dunefsky et al., 2024), as well as pre-activation versus post-activation signals (i.e., features extracted before or after applying the activation function). Table 5 shows that pre-activation features consistently outperform post-activation features for both extractors. Transcoder and SAE achieve similar accuracy, indicating no clear advantage for either architecture. These results suggest that while both extractors are effective, the choice of activation point is more critical, with pre-activations retaining more informative signals about RAG hallucinations.

Table 5: Comparison of SAE and Transcoder feature extractors, using pre- and post-activation signals, for hallucination detection with Llama3.2-1B across three datasets.

| Architecture | Activation | RAGTruth | | AggreFact | | TofuEval | |
|--------------|------------|----------|--------|-----------|--------|----------|--------|
|              |            | Acc      | $F_1$  | Acc       | $F_1$  | Acc      | $F_1$  |
| SAE          | Pre-activation | 0.7810 | 0.7892 | 0.7308 | 0.7388 | 0.6865 | 0.6876 |
|              | Post-activation | 0.7606 | 0.7700 | 0.6939 | 0.7091 | 0.5637 | 0.5642 |
| Transcoder   | Pre-activation | 0.7778 | 0.7830 | 0.7468 | 0.7586 | 0.6652 | 0.6666 |
|              | Post-activation | 0.7594 | 0.7684 | 0.7373 | 0.7525 | 0.6195 | 0.6178 |

**Analysis of Feature Count.** We also examine how varying the number of selected features ($K'$) affects performance, using mutual information (MI) ranking to identify the most informative fea-

tures. Figure 4 shows the performance of Llama2-7B-based RAGLens as $K'$ decreases from 1024 to 1, comparing MI-based selection to random selection (Rand.) starting with the same set of features. As expected, performance drops as fewer features are used, but the decline is much more gradual with MI-based selection, demonstrating that MI effectively prioritizes informative features for hallucination detection. Differences in trends between datasets further highlight the varying complexity of hallucination detection tasks.

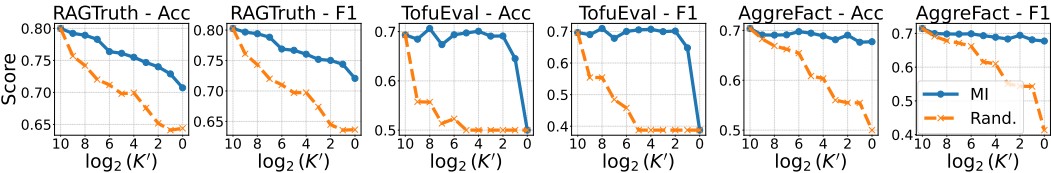

Figure 4: Effect of varying the number of selected features ($K'$) on hallucination detection performance, comparing mutual information (MI) ranking and random selection (Rand.).

**Predictor Ablation.** Lastly, we compare logistic regression (LR), generalized additive model (GAM), multilayer perceptron (MLP), and eXtreme Gradient Boosting (XGBoost) as predictors for hallucination detection using the selected SAE features. Figure 5 shows that GAM consistently outperforms LR and also surpasses more complex models such as MLP and XGBoost, despite its additive structure. This suggests that while the effect of individual features on the output is often nonlinear, the overall contribution of SAE features can be effectively captured in an additive manner. Consequently, GAM is particularly well-suited for leveraging SAE features, offering both strong performance and interpretability.

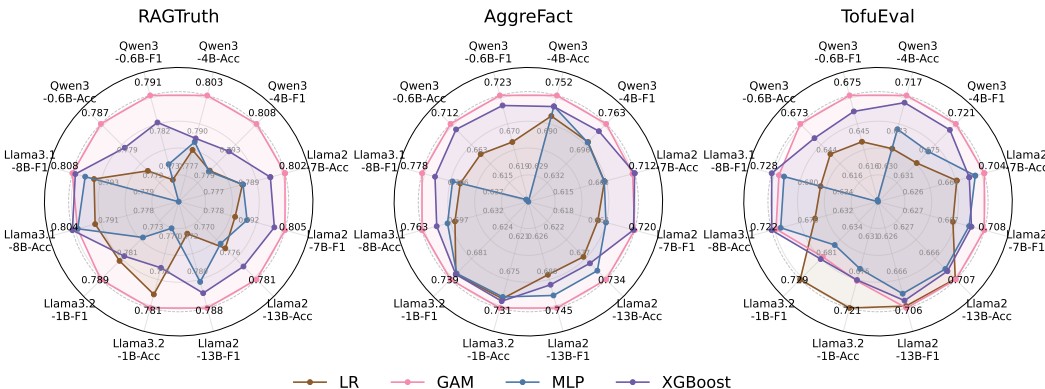

Figure 5: Comparison of logistic regression (LR) and generalized additive model (GAM), multilayer perceptron (MLP), and eXtreme Gradient Boosting (XGBoost) as predictors for RAGLens, evaluated across multiple models and datasets.

## 6 CONCLUSION

In summary, this work demonstrates that SAEs can serve as powerful and interpretable tools for detecting RAG hallucinations. By leveraging internal representations of LLMs, our proposed RAGLens framework not only achieves state-of-the-art performance across multiple benchmarks, but also provides transparent explanations at both local and global levels. Beyond detection, the interpretability of RAGLens enables actionable feedback to mitigate hallucinations, improving the reliability of RAG systems in practice. These findings highlight the broader potential of sparse representation probing for enhancing model faithfulness and open up future directions for integrating lightweight, interpretable SAE-based detectors into real-world applications where trust and accuracy are critical.

ACKNOWLEDGMENTS

This work is supported in part by the US National Science Foundation (NSF) and the National Institute of Health (NIH) under grants IIS-2106913, IIS-2538206, IIS-2529378, CCF-2217071, CNS-2213700, and R01LM014012-01A1. Any recommendations expressed in this material are those of the authors and do not necessarily reflect the views of NIH or NSF.

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

## A    DATASET AND BASELINE DETAILS

### A.1    DATASETS

Here are the detailed descriptions of the datasets used in our experiments.

**RAGTruth.**    RAGTruth (Niu et al., 2024) is a large-scale dataset featuring nearly 18000 naturally generated responses from a range of open- and closed-source LLMs under retrieval-augmented generation settings. The benchmark includes three subtasks: summarization (RAGTruth-Summary), data-to-text generation (RAGTruth-Data2txt), and question answering (RAGTruth-QA). We use the training split of RAGTruth for feature selection and model fitting, and report results on the held-out test set.

**Dolly.**    For the Dolly dataset in the "Accurate Context" setting (Hu et al., 2024), each example presents the model with a context document verified to be relevant and accurate. The model is tasked with generating responses faithful to this context. Following Sun et al. (2025), we perform two-fold cross-validation for model evaluation, alternating between halves of the test set for feature selection and assessment.

**AggreFact.**    AggreFact (Tang et al., 2023) compiles outputs and annotations from various summarization and factual consistency datasets. Our experiments focus on the "SOTA" subset, consisting of summaries produced by models such as T5, BART, and PEGASUS. Each summary is paired with its source document, and factual consistency is annotated by humans. We use the validation set for feature selection and model training, evaluating final performance on the designated test split.

**TofuEval.**    TofuEval (Tang et al., 2024b) is a benchmark designed for topic-focused dialogue summarization. We utilize its MeetingBank portion, which consists of multi-turn meeting transcripts annotated with topic boundaries. LLMs are tasked with generating topic-specific summaries from the full dialogue. Annotations include binary sentence-level factuality as well as free-form explanations for inconsistent content. For this dataset, feature selection and model fitting are performed on the development set, and evaluation is conducted on the test set.

### A.2    BASELINES

We benchmark RAGLens against a diverse set of representative hallucination detection methods. For clarity, we group all baselines into three main categories, detailed below.

**Prompting-based Detectors.**    This category captures hallucination signals by leveraging the generative and reasoning capabilities of LLMs to produce token-level decisions. It encompasses various prompting strategies, such as prompt engineering (Friel & Sanyal, 2023), multi-agent collaboration (Cohen et al., 2023), as well as supervised fine-tuning. Following Sun et al. (2025), we include a fine-tuned Llama2-13B(LR) baseline, where the detector is trained on RAGTruth using LoRA (Hu et al., 2022). The full list of baseline detectors in this category includes: Prompt (Niu et al., 2024), Llama2-13B(LR) (Sun et al., 2025), LMvLM (Cohen et al., 2023), FActScore (Min et al., 2023), FactCC (Kryściński et al., 2020), ChainPoll (Friel & Sanyal, 2023), RAGAS (Es et al., 2024), TruLens (Truera, 2025), RefCheck (Hu et al., 2024), and P(True) (Kadavath et al., 2022).

**Uncertainty-based Detectors.**    These methods assess hallucination likelihood based on the uncertainty of LLM outputs, typically measured via sampled tokens or the distribution of output logits prior to token generation. The complete list of detectors in this category includes: SelfCheckGPT (Manakul et al., 2023), LN-Entropy (Malinin & Gales, 2021), Energy (Liu et al., 2020), Focus (Zhang et al., 2023), and Perplexity (Ren et al., 2023).

**Internal Representation-based Detectors.**    These approaches probe the internal representations of the LLM, such as hidden states, attention patterns, or other intermediate representations, to identify unfaithful generations. By analyzing these internal model dynamics, these detectors aim to capture subtle signals associated with hallucination that may not be reflected in output tokens or

logits. Methods in this category include: EigenScore (Chen et al., 2024), SEP (Han et al., 2024), SAPLMA (Azaria & Mitchell, 2023), ITI (Li et al., 2024), and ReDeEP (Sun et al., 2025).

# B  IMPLEMENTATION DETAILS

While a trained SAE is required for detection in RAGLens, the rapid development of the SAE community (Shu et al., 2025) has led to the public release of many pre-trained SAEs for widely used LLMs. Moreover, SAEs are typically trained on general-purpose corpora rather than task-specific datasets, allowing for their reuse in analyses beyond the scope of this work. To demonstrate the efficiency and practicality of our pipeline, we utilize publicly available SAEs whenever possible, showing that our method does not require resource-intensive, specially tuned SAEs for effective performance. Below, we list the specific SAEs used in our experiments:

- **Llama2-7B:** We use the pretrained SAE from `https://huggingface.co/yuzhaouoe/Llama2-7b-SAE`, which includes SAEs for multiple layers. Specifically, we select the SAE trained on "layers.15", with an expansion factor of 32 and Top-K activation ($K = 192$).

- **Llama2-13B:** As no public SAE is available for Llama2-13B, we train our own using the `sparsify` package[1] with default settings. The SAE is trained on "layers.15", with an expansion factor of 16 and Top-K activation ($K = 16$).

- **Llama3.2-1B:** We use the pretrained SAE from `https://huggingface.co/EleutherAI/sae-Llama-3.2-1B-131k`, which covers multiple layers. For results in Section 4.3, we select the SAE trained on "layers.6.mlp", and for the layer-wise analysis in Section 5, we use all available SAEs. The SAEs have an expansion factor of 32 and Top-K activation ($K = 32$).

- **Llama3-8B:** We use the pretrained SAE from `https://huggingface.co/EleutherAI/sae-llama-3-8b-32x`, utilizing all available SAEs for the layer-wise analysis in Section 5. The SAEs have an expansion factor of 32 and Top-K activation ($K = 192$).

- **Llama3.1-8B:** We use the pretrained SAE from `https://huggingface.co/Goodfire/Llama-3.1-8B-Instruct-SAE-l19`, which contains the SAE trained on "layers.19". The SAE has an expansion factor of 16 and ReLU activation.

- **Qwen3-0.6B:** As there is no public SAE for Qwen3-0.6B, we train our own using the `sparsify` package with default settings. For results in Section 4.3, we select the SAE trained on "layers.17", and for layer-wise analysis, we use all trained SAEs. The SAEs have an expansion factor of 32 and Top-K activation ($K = 16$).

- **Qwen3-4B:** Similarly, we train our own SAEs for Qwen3-4B. For Section 4.3, we select the SAE trained on "layers.22"; for layer-wise analysis, we use all available SAEs. The SAEs have an expansion factor of 32 and Top-K activation ($K = 16$).

To compute mutual information (MI) in RAGLens, we estimate the MI value of each continuous SAE feature by discretizing feature values into bins using quantile thresholds. Specifically, we partition the value range into 50 bins per feature and compute MI values in chunks for GPU acceleration. After ranking features based on estimated MI, we select the top 1000 features for subsequent Generalized Additive Model (GAM) fitting in RAGLens.

For GAM fitting, we deploy the Explainable Boosting Machine (EBM) (Nori et al., 2019), a high-performance, tree-based GAM implementation that flexibly models nonlinear effects of features on the target output. Across all experiments, we set the maximum number of bins in each feature's shape function to 32, a validation size of $10\%$, and a maximum of 1000 boosting rounds.

To generate semantic explanations for selected SAE features, we collect representative activation cases by sampling 12 examples with the highest activations and 12 examples distributed across quantiles from the RAGTruth training data. These cases are then provided to GPT-5, which summarizes the underlying semantic concept captured by each feature using the template in Figure 7.

---

[1]`https://github.com/EleutherAI/sparsify`

Prompt templates for all LLM text-generation calls are shown in Appendix J.

All experiments are conducted on a server equipped with an AMD EPYC 7313 CPU and four NVIDIA A100 GPUs.

## C CAUSAL INTERVENTION OF SAE FEATURES

For SAE features that consistently activate prior to hallucinated content, we investigate whether direct intervention on these features can causally influence the model's generation. Specifically, we manipulate the activation values of selected SAE features at key tokens preceding hallucinated spans and observe the resulting changes in model outputs. This analysis assesses whether these features not only correlate with hallucination but also play a causal role in driving unfaithful generations.

Table 6 presents a case study on Feature 22790 from Llama3.1-8B, which reliably activates before hallucinated numeric or temporal details (e.g., firing on the token "of" in the prefix "[...] at the age of", which often leads to unsupported ages). When we suppress this feature (e.g., set its value to 0 or -20), the model continues the problematic prefix with hallucinated, ungrounded numbers. In contrast, manually setting the feature to a large positive value (e.g., 20) steers the model to remain faithful to the context, producing follow-up tokens that avoid hallucinated specifics (e.g., using an unspecified age or time). This suggests that Feature 22790 reflects the model's awareness of potentially hallucinated numeric or temporal details, and that overactivating it can encourage more faithful behavior in such scenarios.

Table 6: Examples of causal interventions on Feature 22790 in Llama3.1-8B. This feature is consistently activated prior to hallucinated numeric/time specifics. Tokens in red indicate the hallucinated content.

| Context | Prefix | Value | Output |
|---------|--------|-------|--------|
| No mention of the age | *[...] at the age of* | -20.0 | *[...] of 30.* |
| | | 0.0 | *[...] of 25.* |
| | | 20.0 | *[...] of an unspecified age.* |
| No mention of the release date | *[...] scheduled to be released in* | -20.0 | *[...] in 2016.* |
| | | 0.0 | *[...] in the future.* |
| | | 20.0 | *[...] in an unspecified time frame.* |

An additional case study on Feature 71784 in Llama2-13B is shown in Table 7. This feature is associated with hallucinations about opening hours (day/time) and ratings, and typically activates only when the hallucinated time or rating is already being produced (e.g., "Monday" in an inconsistent hours-of-operation statement). For these cases, we perturb the feature value on the token immediately preceding the hallucinated word (e.g., "on" in "on Mondays"). The results also confirm that manipulating SAE features identified by RAGLens can steer the model's behavior and demonstrate a causal relationship between these features and RAG hallucinations.

However, for features like the one in Table 7 that only activate concurrently with or after hallucinated tokens, direct intervention is impractical for preventing hallucinations, as the problematic content has already been generated by the time these features fire. Furthermore, the distance between the hallucinated tokens and the token with high activation is not always consistent across features and examples. Thus, while causal intervention on SAE features is feasible in certain scenarios, it is not a universal solution for hallucination mitigation. This limitation motivates our focus on post-hoc text-based feedback in the main mitigation pipeline.

## D DISCUSSION ON SAE FEATURE VERSUS HIDDEN STATE

To further investigate the contribution of SAE-derived features to RAGLens performance, we conduct an ablation study by replacing SAE features with the raw hidden states of Llama2-7B, while

Table 7: Examples of causal interventions on Feature 71784 in Llama2-13B. This feature is consistently activated on hallucinations about opening hours (day/time) and ratings. Tokens in red indicate the hallucinated content.

| Context | Prefix | Value | Output |
|---------|--------|-------|--------|
| *[...] "Monday": "0:0-0:0" [...]* | *[...] Restaurants Hours:* | -20.0 | *[...]: the restaurant is open from 11:00 AM to 21:00 PM, Monday to Sunday.* |
| | | 0.0 | *[...]: the restaurant is open from 11:00 AM to 21:00 PM, Monday to Sunday.* |
| | | 20.0 | *[...]: Monday: 0:00 - 0:00 [...]* |
| *[...] "Monday": "0:0-0:0" [...]* | *[...] no information available for the business's hours on* | -20.0 | *[...] on hours of operation on Mondays.* |
| | | 0.0 | *[...] on hours of operation on Mondays.* |
| | | 20.0 | *[...] on holidays.* |

retaining the MI-based feature selection and GAM classifier. Figure 6 compares hallucination detection performance using hidden states versus SAE features across varying numbers of selected dimensions ($K'$). When $K'$ is large, hidden states achieve performance comparable to SAE features, which is expected since SAE features are derived from hidden states.

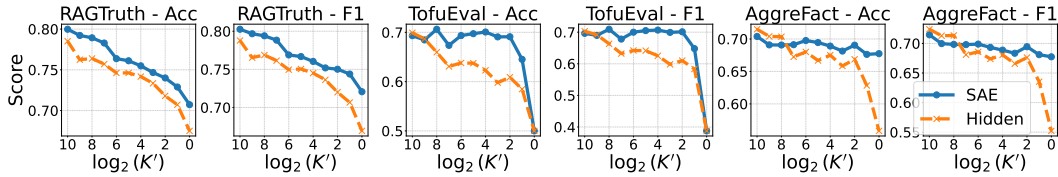

Figure 6: Effect of varying the number of selected features ($K'$) on hallucination detection performance, comparing SAE features (SAE) and hidden states (Hidden).

However, as $K'$ decreases, the performance of hidden states degrades more rapidly, especially on single-task datasets such as AggreFact and TofuEval. This indicates that SAE features more effectively disentangle hallucination-related signals from other information and concentrate them into a compact set of dimensions, particularly when the predictor is applied to narrow domains. Such compact representations improve interpretability and facilitate downstream applications like hallucination mitigation, as only a small number of salient features need to be monitored to achieve effective control.

# E   DISCUSSION ON SAE FEATURE INTERPRETABILITY

To validate the semantic consistency of SAE features, we extend the analysis in Table 3 by examining top-activated examples from the SAE's pretraining corpus. Specifically, we compute activations for Feature 22790 in Llama3.1 8B over the first 10,000 samples from the lmsys/lmsys chat 1m corpus[2] and inspect the highest activation cases, as shown in Table 8. Although the pretraining corpus contains diverse structures and languages, including clinical dialogue transcripts and telecom statements in German, we find that Feature 22790 is consistently activated in scenarios where the response is about to produce specific numbers or dates that are likely hallucinated. This pattern aligns with the feature's summarized semantic meaning from RAGTruth and the representative examples shown in Table 3, demonstrating that the feature robustly captures hallucination-related signals across both task-specific and pretraining data.

To further assess the robustness of the distilled feature explanation across diverse scenarios, we prompt GPT-5 to predict the activation level of Feature 22790 on 24 held-out RAGTruth test cases

---

[2]https://huggingface.co/datasets/lmsys/lmsys-chat-1m

Table 8: Examples from the pre-training corpus with high activations of Feature 22790 in Llama3.1-8B. Tokens highlighted in red indicate locations of strong feature activation.

| Sample Index | Input | Output |
|---|---|---|
| 9384 | *TRANSCRIPT='Hello doctor I have fever and cough. Okay take paracetamol and go home and rest.' [...]* | *[...] Patient advises they have been experiencing symptoms for the past two days [...]* |
| 3298 | *User: Telekom Deutschland GmbH [...]* | *[...] "summary": "Mobilfunk-Rechnung für den Monat März 2023" [...]* |

(comprising 8 top-activated and 16 quantile-distributed examples), using only the natural-language summary. For each case, the three most highly activated tokens are highlighted, and GPT-5 is asked to rate the expected feature activation on a scale from 0 (feature not present) to 5 (very strong match). Comparing these predicted scores to the actual SAE activations yields a Pearson correlation of 0.6731 ($p < 0.05$), suggesting that the explanation reliably captures when the feature should activate across a broad range of examples.

However, while SAEs are designed to disentangle distinct semantic concepts from hidden states, some SAE features may remain generic or polysemantic, limiting their interpretability, which is a challenge widely recognized in current SAE research (Bricken et al., 2023; Huben et al., 2024). Although RAGLens already achieves accurate and interpretable RAG hallucination detection using existing SAE foundations, its architecture-agnostic design allows it to benefit from future advances in SAE methods, which may enable even more transparent and effective detectors for RAG hallucinations.

## F ABLATION STUDY OF SAE FEATURE EXTRACTION METHODS IN RAGLENS

In this section, we investigate whether our SAE-based design, which combines max pooling over tokens with mutual-information-based feature selection, offers advantages over alternative ways of using SAE features (Ferrando et al., 2025; Tillman & Mossing, 2025; Xin et al., 2025; Suresh et al., 2025). Concretely, we compare four variants on the RAGTruth subtasks:

1. **Last Token + Selection + GAM**: SAE features taken only from the last generated token, followed by MI-based feature selection and a GAM classifier.

2. **Max Pooled + No Selection + LR**: max-pooled SAE features across tokens, using all dimensions as input to a logistic regression (LR) classifier (no feature selection).

3. **Max Pooled + Selection + LR**: max-pooled SAE features with MI-based feature selection, using LR as the classifier.

4. **Max Pooled + Selection + GAM (RAGLens)**: our full RAGLens variant, which applies MI-based feature selection to max-pooled SAE features and then fits a GAM.

Table 9 reports accuracy and AUC for these settings. Using max pooling with feature selection and a GAM (*Max Pooled + Selection + GAM*) consistently outperforms both: (i) the last-token baseline (*Last Token + Selection + GAM*), which discards earlier activations that may precede hallucinated content, and (ii) the *Max Pooled + No Selection + LR* baseline, which uses all SAE dimensions without selection and is thus less efficient.

Additionally, *Max Pooled + Selection + LR* performs comparably to using all features without selection, indicating that MI-based selection preserves the most informative SAE dimensions for detecting unfaithful model outputs. Overall, these results demonstrate that our approach captures most hallucination-relevant signals in a compact, efficient feature set and achieves superior detection performance compared to alternative SAE usage strategies.

Table 9: Ablation study of SAE feature extraction and classifier choices on RAGTruth subtasks using Llama2-7B. We compare last-token versus max-pooled SAE features, with and without mutual information-based feature selection. Due to the high computational cost of fitting GAMs on all features, we use logistic regression (LR) as the classifier when no feature selection is applied.

| Source | Selection | Classifier | RAGTruth-Summary | | RAGTruth-QA | | RAGTruth-Data2txt | |
|--------|-----------|-----------|------|------|------|------|------|------|
| | | | Acc | AUC | Acc | AUC | Acc | AUC |
| Last Token | Yes | GAM | 0.6293 | 0.7507 | 0.6908 | 0.8101 | 0.7454 | 0.8296 |
| Max Pooled | No | LR | 0.6734 | 0.7305 | 0.7356 | 0.8344 | 0.7499 | 0.8397 |
| Max Pooled | Yes | LR | 0.6718 | 0.7663 | 0.7572 | 0.8472 | 0.7085 | 0.8014 |
| Max Pooled | Yes | GAM | 0.6973 | 0.8191 | 0.7717 | 0.8835 | 0.7668 | 0.8454 |

# G   PROOF OF THEOREM 1

**Theorem 2** (Restatement of Theorem 1). *Fix a feature index $k$ and suppress $k$ in notation. For tokens $t = 1, \ldots, T$,*

$$z_t = \begin{cases} 0 & \text{with probability } 1 - p_\ell, \\ V_t & \text{with probability } p_\ell, \end{cases} \quad \text{independently over } t, \tag{10}$$

*where $V_t$ are i.i.d. from a label-independent distribution $F$ on $(0, \infty)$. Let $\bar{z} = \max_{1 \leq t \leq T} z_t$, $\pi = \Pr(\ell = 1)$, $\bar{p} = \frac{1}{2}(p_1 + p_0)$, and $\Delta p = p_1 - p_0$. If $T\bar{p} \ll 1$, then (in bits)*

$$I(\bar{z}; \ell) = \frac{\pi(1-\pi)}{2 \ln 2} \frac{T(\Delta p)^2}{\bar{p}} + O\big((T\bar{p})^2\big). \tag{11}$$

*In particular, $I(\bar{z}; \ell) > 0$ iff $p_1 \neq p_0$; the leading dependence is linear in $T$ and quadratic in $\Delta p$.*

*Proof.* Write $A = \mathbf{1}\{\bar{z} > 0\}$, $N = \sum_{t=1}^{T} \mathbf{1}\{z_t > 0\}$, and $h(u) = -u \log_2 u - (1-u) \log_2(1-u)$.

**Step 1: Exact MI for the activation indicator.**   Independence across tokens implies

$$q_\ell := \Pr(A=1 \mid \ell) = \Pr(N \geq 1 \mid \ell) = 1 - (1 - p_\ell)^T. \tag{12}$$

Hence $A \mid \ell \sim \text{Bernoulli}(q_\ell)$ and

$$I(A; \ell) = h\big(\pi q_1 + (1-\pi)q_0\big) - \big[\pi\, h(q_1) + (1-\pi)\, h(q_0)\big]. \tag{13}$$

**Step 2: Small-gap expansion of $I(A; \ell)$.**   Let $r = \pi q_1 + (1-\pi)q_0$, $\bar{q} = \frac{1}{2}(q_1 + q_0)$, and $\Delta q = q_1 - q_0$. A second-order Taylor expansion of $h$ about $\bar{q}$ gives

$$h(q_1) = h(\bar{q}) + \frac{\Delta q}{2} h'(\bar{q}) + \frac{(\Delta q)^2}{8} h''(\bar{q}) + O((\Delta q)^3), \tag{14}$$

$$h(q_0) = h(\bar{q}) - \frac{\Delta q}{2} h'(\bar{q}) + \frac{(\Delta q)^2}{8} h''(\bar{q}) + O((\Delta q)^3), \tag{15}$$

$$h(r) = h(\bar{q}) + (2\pi - 1)\frac{\Delta q}{2} h'(\bar{q}) + \frac{(2\pi-1)^2(\Delta q)^2}{8} h''(\bar{q}) + O((\Delta q)^3). \tag{16}$$

Plugging into Equation 13, the linear terms cancel, and since $h''(u) = -[u(1-u)\ln 2]^{-1}$,

$$I(A; \ell) = \frac{\pi(1-\pi)}{2 \ln 2} \frac{(\Delta q)^2}{\bar{q}(1-\bar{q})} + O((\Delta q)^3). \tag{17}$$

**Step 3: Relating $I(\bar{z}; \ell)$ and $I(A; \ell)$, with a JS–TV bound.**   Because $A = \mathbf{1}\{\bar{z} > 0\}$ is a deterministic function of $\bar{z}$, the chain rule gives

$$I(\bar{z}; \ell) = I(A; \ell) + I(\bar{z}; \ell \mid A). \tag{18}$$

When $A = 0$, $\bar{z} = 0$ almost surely, so $I(\bar{z}; \ell \mid A = 0) = 0$. When $A = 1$, one can write

$$p_{\bar{z}|A=1,\ell} \;=\; \sum_{n \geq 1} w_\ell(n)\, F^{(n)}, \tag{19}$$

where $w_\ell(n) = \Pr(N = n \mid A = 1, \ell)$ and $F^{(n)}$ is the distribution of the maximum of $n$ label-independent draws from $F$. Thus given $A = 1$, the only dependence on $\ell$ is via the mixture weights $\{w_1(n) - w_0(n)\}$.

By the definition of total variation (TV) distance,

$$\begin{aligned}
\mathrm{TV}\big(p_{\bar{z}|A=1,1},\, p_{\bar{z}|A=1,0}\big) \;&=\; \tfrac{1}{2} \int \big|p_{\bar{z}|A=1,1}(z) - p_{\bar{z}|A=1,0}(z)\big|\, dz \\
&\leq\; \Pr(N \geq 2 \mid A = 1, 1) + \Pr(N \geq 2 \mid A = 1, 0),
\end{aligned} \tag{20}$$

up to constant factors.

Since mutual information conditioned on $A = 1$ is the Jensen–Shannon (JS) divergence between these two conditional distributions (with weight $\Pr(\ell = 1 \mid A = 1)$ over $\ell$), one can invoke a standard bound:

$$\mathrm{JS}\big(p_{\bar{z}|A=1,1},\, p_{\bar{z}|A=1,0}\big) \;\leq\; C\, \mathrm{TV}\big(p_{\bar{z}|A=1,1},\, p_{\bar{z}|A=1,0}\big)^2, \tag{21}$$

for some constant $C$ depending on the mixing weight (Corander et al., 2021).

Thus,

$$I(\bar{z}; \ell \mid A = 1) \;=\; O\big((\Pr(N \geq 2 \mid A = 1, 1) + \Pr(N \geq 2 \mid A = 1, 0))^2\big). \tag{22}$$

Multiplying by $\Pr(A = 1) \leq 1$ gives

$$I(\bar{z}; \ell) - I(A; \ell) \;=\; O\big(\Pr(N \geq 2 \mid 1)^2 + \Pr(N \geq 2 \mid 0)^2\big), \tag{23}$$

which, under rarity ($T\bar{p} \ll 1$), is $o((T\bar{p})^2)$.

**Step 4: Specializing to independence and the sparse regime.** Under independence with per-token rate $p_\ell$,

$$\Pr(N \geq 2 \mid \ell) \;=\; 1 - (1 - p_\ell)^T - Tp_\ell(1 - p_\ell)^{T-1} \;=\; \binom{T}{2} p_\ell^2 + O(T^3 p_\ell^3). \tag{24}$$

Thus

$$I(\bar{z}; \ell) \;=\; I(A; \ell) \;+\; O\big(T^2 \bar{p}^2\big), \tag{25}$$

uniformly for $p_\ell$ with $\bar{p} = \tfrac{1}{2}(p_1 + p_0)$.

**Step 5: Substitute sparse approximations.** For $T\bar{p} \ll 1$,

$$q_\ell = 1 - (1 - p_\ell)^T = Tp_\ell - \binom{T}{2} p_\ell^2 + O(T^3 p_\ell^3), \tag{26}$$

$$\bar{q} = T\bar{p} + O(T^2 \bar{p}^2), \tag{27}$$

$$\Delta q = T\Delta p + O(T^2 \bar{p}\,|\Delta p|). \tag{28}$$

Plug these into Equation 17. Since $1 - \bar{q} = 1 + O(T\bar{p})$,

$$\frac{(\Delta q)^2}{\bar{q}(1 - \bar{q})} = \frac{T^2 (\Delta p)^2}{T\bar{p}} \;+\; O\big(T^2 (\Delta p)^2 \cdot T\bar{p}\big) = \frac{T(\Delta p)^2}{\bar{p}} \;+\; O\big(T^3 \bar{p}\,(\Delta p)^2\big). \tag{29}$$

Moreover, $(\Delta q)^3 = O\big(T^3 |\Delta p|^3\big) = o\big(T^2 \bar{p}^2\big)$ under $T\bar{p} \ll 1$ and bounded $|\Delta p|$. Hence

$$I(A; \ell) = \frac{\pi(1 - \pi)}{2\ln 2}\, \frac{T(\Delta p)^2}{\bar{p}} \;+\; O\big(T^2 \bar{p}^2\big). \tag{30}$$

Combining Equation 30 with Equation 25 via Equation 18 yields

$$I(\bar{z}; \ell) = \frac{\pi(1 - \pi)}{2\ln 2}\, \frac{T(\Delta p)^2}{\bar{p}} \;+\; O\big(T^2 \bar{p}^2\big), \tag{31}$$

which is the claimed statement since $T^2 \bar{p}^2 = (T\bar{p})^2$. $\qquad\square$

## H  INTERPRETATION OF ADDITIONAL HALLUCINATION-RELATED FEATURES

Table 10 highlights additional representative features that RAGLens activates when detecting hallucinations across LLMs of varying scales. These features capture a wide range of hallucination patterns, including overconfident numeric spans, incorrect temporal assertions, and ungrounded entity mentions. Notably, smaller LLMs tend to exhibit more generic hallucination signals (e.g., "overstated concrete details"), whereas larger LLMs reveal more specific and nuanced patterns (e.g., "precise numeric spans not grounded in retrieved passages"). This observation suggests that as LLMs increase in size, they develop more specialized internal features for identifying complex hallucination phenomena. This may contribute to the improved hallucination detection performance of larger models with RAGLens, as shown in Figure 2.

Table 10: Explanation and examples of representative SAE features in Llama2-7B, Llama2-13B, Llama3.2-1B, and Llama3.1-8B that enable interpretable hallucination detection. Spans highlighted in red indicate tokens where the feature is highly activated. The table illustrates how these features capture diverse hallucination patterns across models.

| Feature ID | Feature Explanation | Example Output | Example Explanation |
|---|---|---|---|
| | | Llama2-7B | |
| 120059 | hallucination involving entity swaps and invented details. | [...] his castmates from "Central Intelligence" took a knee as he [...] | There is no mention of Kevin Hart's castmates from "Central Intelligence". |
| 127083 | unsupported concrete additions such as names, pairings, numbers, legal / evidentiary claims | [...] such as a Walker-Rubio or Clinton-Kaine pairing [...] | Clinton-Kaine is not mentioned in the source content |
| | | Llama2-13B | |
| 26530 | "amenity assertion" detector that spikes on the outdoor seating phrase | The restaurant has a casual ambiance and offers outdoor seating | Original text shows no outdoor seating |
| 71784 | hallucinations on day/-time (hours) and ratings | [...] no information available for the business's hours on Mondays [...] | Original text: closed on Monday |
| | | Llama3.2-1B | |
| 78162 | unsupported or swapped named entities and precise facts | The team is now facing their in-state rivals, the Los Angeles Dodgers [...] | The Texas Rangers and Los Angeles Dodgers are not in-state rivals |
| 121247 | invented or overstated concrete details | [...] He was also a professor of film criticism at NYU [...] | It is not mentioned in the original source. |
| | | Llama3.1-8B | |
| 37877 | precise numeric spans that aren't grounded in the retrieved passages | [...] soft ball stage occurs at a temperature of around 245-250°Fahrenheit [...] | The firm ball stage at a temperature of about 245 to 250 degrees Fahrenheit |
| 40779 | overconfident claims about hours, open/closed days, and amenities | [...] the exception of Friday when it closes at 20:00 PM | Friday opens from 11am and closes by 10pm |

## I   FURTHER ANALYSIS OF IDENTIFIED FEATURES VIA COUNTERFACTUAL PERTURBATION

To further validate that the features identified by RAGLens capture meaningful hallucination patterns in RAG-specific contexts, we conduct case studies using counterfactual perturbation on SAE feature 37877 from Llama3.1-8B, which detects "precise numeric spans not grounded in retrieved passages" (see Table 10). We select representative samples from three RAGTruth subtasks, summarization, data-to-text, and question answering, where this feature is highly activated and the generation is hallucinated. For each, we manually edit the context to construct counterfactual scenarios: (1) the output becomes consistent with the perturbed context, or (2) the output remains inconsistent, but in a different way. For question answering, we also consider a version where the context is entirely removed to examine feature activation in the absence of grounding.

Tables 11-13 present the results. For summarization (Table 11) and data-to-text (Table 13), when the context is edited to make the output consistent, the feature value on previously highlighted tokens drops significantly; if the context remains inconsistent, the feature stays highly activated. In question answering (Table 12), feature activation drops when the context is either edited to be consistent or removed, indicating that the feature is specialized for detecting ungrounded numeric spans in context, rather than general hallucination. Overall, these case studies demonstrate that RAGLens identifies features that robustly capture RAG-specific hallucination patterns.

## J   PROMPT TEMPLATES FOR LLM CALLING

This section presents the prompt templates we use for LLM calling in our experiments. Specifically, for the summarized SAE explanations in Table 3, we use the template shown in Figure 7. The hallucination mitigation approaches discussed in Section 4.6 are implemented with templates in Figures 8 and 9 for the instance- and token-level feedback, respectively. The LLM evaluation shown in 4 is implemented with the template in Figure 10, following Luo et al. (2023).

---

**Prompt template for summarizing SAE feature explanations**

I am trying to explain the semantic meaning of a hallucination-related feature in retrieval-augmented generation settings. Please first examine if the highly activated tokens in each example are related to specific cases of hallucinations. Then, try to summarize the semantic meaning of the feature based on these observations. Finally, give me a concise description of the feature meaning in one sentence, specifying what kind of hallucination (if applicable) it is detecting.

### Example

#### Here is the input:
{{input}}

#### Here is the output:
{{output}}

#### Here are the feature activation associated with each output token:
{{[(token1, value1), (token2, value2), ...]}}

### Example
...
...

---

Figure 7: Prompt template for summarizing SAE feature explanations.

Table 11: Counterfactual analysis of SAE feature 37877 from Llama3.1-8B on a RAGTruth-Summary example. The table illustrates how feature activation changes in response to context perturbations, highlighting its specificity for detecting ungrounded numeric spans in summarization. Key information in the original and updated contexts is highlighted in yellow. For the output, tokens are highlighted in red, with the intensity of the red background indicating the relative activation value on each token.

| Model Input |
| --- |
| Summarize the following news within 78 words: Iraq Service Campaign Medal Reinstated [...] According to a memorandum signed by then-Under Secretary of Defense for Personnel and Readiness Gilbert Cisneros in August, the Pentagon once again awarded the Inherent Resolve Campaign Medal to eligible service members serving in Iraq. According to a document received by Military Times, the medal's eligibility has been renewed retroactive to January 1 and will continue through December 31, 2024. [...] |

| Model Output (annotated with feature activation) |
| --- |
| The US Department of Defense has reinstated the Inherent Resolve Campaign Medal for service members in Iraq due to renewed threats against US personnel in the Middle East. The medal's eligibility has been renewed retroactively from January 1, 2021, and will continue through December 31, 2024. The move was requested by the US Central Command, following Iraq's reclassification from a combat mission. The medal was initially created in 2016 to acknowledge service in Iraq or Syria, but was limited to Syria in 2017. |

| Context Perturbation 1 (context inconsistent) |
| --- |
| **Original Context**: [...] the medal's eligibility has been renewed retroactive to January 1 [...] |
| **Updated Context**: [...] the medal's eligibility has been renewed retroactive to January 1, 2023 [...] |
| **Updated Activation**: [...] The medal's eligibility has been renewed retroactively from January 1, 2021, and will continue through December 31, 2024 [...] |

| Context Perturbation 2 (context consistent) |
| --- |
| **Original Context**: [...] the medal's eligibility has been renewed retroactive to January 1 [...] |
| **Updated Context**: [...] the medal's eligibility has been renewed retroactive to January 1, 2021 [...] |
| **Updated Activation**: [...] The medal's eligibility has been renewed retroactively from January 1, 2021, and will continue through December 31, 2024 [...] |

## K USE OF LARGE LANGUAGE MODELS

In this project, large language models (LLMs) are used for multiple purposes:

- We use checkpoints of open-source LLMs to extract hidden states, which are then analyzed to identify interpretable SAE features that help detect RAG hallucinations.
- We use LLMs as judges to evaluate whether RAGLens feedback helps mitigate hallucinations in the original model outputs.
- We use LLMs as summarizers to describe the semantic roles of different SAE features based on their activations across multiple samples.
- We use LLMs to proofread the paper.

---

**Prompt template for hallucination mitigation with instance-level feedback**

User:
`{{input}}`

Assistant:
`{{original_output}}`

User:
There are hallucinations in your output. Please revise it.

---

Figure 8: Prompt template for hallucination mitigation with instance-level feedback.

---

**Prompt template for hallucination mitigation with token-level feedback**

User:
`{{input}}`

Assistant:
`{{original_output}}`

User:
There are hallucinations in your output, especially on the following spans:
`{{[span1, span2, ...]}}`

Please revise it.

---

Figure 9: Prompt template for hallucination mitigation with token-level feedback.

---

**Prompt template for LLM-as-a-Judge on mitigation results**

Decide if the following summary/answer is consistent with the corresponding article. Note that consistency means all information in the output is supported by the article.

Article: `{{context}}`

Summary/Answer: `{{revised_output}}`

Explain your reasoning step by step then answer (yes or no) the question:

---

Figure 10: Prompt template for LLM-as-a-Judge on mitigation results.

Table 12: Counterfactual analysis of SAE feature 37877 from Llama3.1-8B on a RAGTruth-QA example. The table illustrates how feature activation changes in response to context removal and consistency edits, highlighting its sensitivity to ungrounded numeric spans in retrieval-augmented question answering. Key information in the original and updated contexts is highlighted in yellow. For the output, tokens are highlighted in red, with the intensity of the red background indicating the relative activation value on each token.

| **Model Input** |
| --- |
| Briefly answer the following question:
how to tell the temperature of water with a candy thermometer

Bear in mind that your response should be strictly based on the following three passages:
passage 1: [...] If the candy forms a soft, flexible ball it is at a temperature of 235-240 degrees Fahrenheit or at the soft ball stage.
passage 2: [...] If it forms into a firm ball, it is in the firm ball stage. This stage is at a temperature of about 245 to 250 degrees Fahrenheit [...]
passage 3: [...]

In case the passages do not contain the necessary information to answer the question, please reply with: "Unable to answer based on given passages." |
| **Model Output (annotated with feature activation)** |
| Based on the provided passages, here is how to tell the temperature of water using a candy thermometer:
Passage 1 states that when the candy is in the thread stage, the temperature is around 230-235°Fahrenheit.
Passage 2 indicates that the soft ball stage occurs at a temperature of 245-250°Fahrenheit for candies such as fudge and fondant.
Therefore, to determine the temperature of water using a candy thermometer, you can follow these steps:
1. Cool some of the candy in the thread stage by dropping it into cold water.
2. Observe if the candy forms a soft, flexible ball. If it does, the temperature is between 245-250°Fahrenheit, indicating that the water is at the soft ball stage.
If the water is not at the soft ball stage, continue cooling it until it reaches that temperature range, and then proceed to use the candy thermometer to measure its temperature. |
| **Context Perturbation 1 (context removed)** |
| **Original Context**:
[...] Bear in mind that [...]

**Updated Context**:
None

**Updated Activation**:
[...] indicates that the soft ball stage occurs at a temperature of around 245-250°Fahrenheit [...] |
| **Context Perturbation 2 (context consistent)** |
| **Original Context**:
[...] This stage is at a temperature of about 245 to 250 degrees Fahrenheit [...]

**Updated Context**:
[...] The soft ball stage is at a temperature of about 245 to 250 degrees Fahrenheit [...]

**Updated Activation**:
[...] indicates that the soft ball stage occurs at a temperature of around 245-250°Fahrenheit [...] |

Table 13: Counterfactual analysis of SAE feature 37877 from Llama3.1-8B on a RAGTruth-Data2txt example. The table demonstrates how feature activation responds to context perturbations, highlighting its specificity for identifying ungrounded numeric spans in data-to-text generation. Key information in the original and updated contexts is highlighted in yellow. For the output, tokens are highlighted in red, with the intensity of the red background indicating the relative activation value on each token.

| Model Input |
| --- |
| Write an objective overview about the following local business based only on the provided structured data in the JSON format. You should include details and cover the information mentioned in the customers' review. The overview should be 100 - 200 words. Don't make up information. Structured data: {'name': "Mony's Mexican Food", 'address': '217 E Anacapa St', 'city': 'Santa Barbara', 'state': 'CA', 'categories': 'Restaurants, Mexican', 'hours': {'Monday': '10:30-15:30', 'Tuesday': '10:30-15:30', 'Wednesday': '10:30-15:30', 'Thursday': '10:30-15:30', 'Friday': '10:30-15:30', 'Saturday': '10:30-15:30'}, [...]} |

| Model Output (annotated with feature activation) |
| --- |
| Sure! Here's an objective overview of Mony's Mexican Food based on the provided structured data: 

 [...] The restaurant has a cozy and casual atmosphere, with no WiFi or outdoor seating available. However, customers can enjoy takeout or dine in during its hours of operation, which are from 10:30 AM to 3:30 PM from Monday to Friday, and 10:30 AM to 4:00 PM on Saturday. Despite the limited hours and lack of WiFi, Mony's Mexican Food remains a popular choice among locals and visitors alike, with many return customers and high praise from reviewers. |

| Context Perturbation 1 (context inconsistent) |
| --- |
| **Original Context**: 
 [...] 'Saturday': '10:30-15:30'[...] 

 **Updated Context**: 
 [...] 'Saturday': '9:30-15:30' [...] 

 **Updated Activation**: 
 [...] and 10:30 AM to 4:00 PM on Saturday [...] |

| Context Perturbation 2 (context consistent) |
| --- |
| **Original Context**: 
 [...] 'Saturday': '10:30-15:30' [...] 

 **Updated Context**: 
 [...] 'Saturday': '10:30-16:00' [...] 

 **Updated Activation**: 
 [...] and 10:30 AM to 4:00 PM on Saturday [...] |

