# OpenReview forum: "Toward Faithful Retrieval-Augmented Generation with Sparse Autoencoders"
_ICLR.cc/2026/Conference — ICLR 2026 Poster_

### Official Review · Reviewer_sTa7 · 2025-10-28

**Soundness:** 3
**Presentation:** 3
**Contribution:** 2
**Rating:** 4
**Confidence:** 4

**Summary:**

This paper proposes a pipeline dubbed “RAGLens” for detecting hallucinations in retrieval augmented generation (RAG) outputs of LLMs. Specifically, the authors employ sparse autoencoders (SAEs) to extract interpretable and hallucination-relevant internal LLM features in RAG outputs. Key SAE features are selected based on the mutual information (MI) between the max-pooled token features and the hallucination label, and a generalized additive model (GAM) is fitted on these selected features to predict the hallucination label. Hallucination detection results from RAGLens are compared with existing hallucination detection baselines, and a hallucination mitigation strategy using the results of RAGLens is proposed.

**Verdict**: This paper is clearly presented and empirically solid, but only a mildly novel application of known SAE-based hallucination detection techniques to RAG outputs. The problem space and proposed method do not seem particularly unique and well motivated, and therefore I do not recommend acceptance at this stage. However, the authors could convince me otherwise by addressing each of the major weaknesses below.

**Strengths:**

S1: The paper is mostly well-written and the methodology for the MI feature selection and GAM implementation is well laid out and explained.
S2: A wide array of baseline methods of different varieties are used for comparison, as well as a good number of evaluation datasets. These results help to bolster the credibility and apparent utility of the method.

**Weaknesses:**

Major:
- W1: While the paper is quite thorough in the method presented, the main idea of detecting hallucinations in RAG or grounded outputs using SAEs is of limited novelty. There are many papers showing that SAE features can be used for hallucination detection in generated outputs [1][2][3][4], and in fact there have been papers that show this for RAG outputs as well (including mitigation) [5]. While the proposed method is clearly effective, it is not properly motivated as to why RAG hallucinations should be treated differently from generic LLM hallucinations. Indeed the authors state “However, it remains unclear whether such features can reliably signal RAG-specific hallucination modes, such as evidence conflicts or unsupported novel content”. But they do not motivate why RAG outputs should present a different paradigm from generic hallucinated outputs. Some of the aforementioned works should be cited and contrasted with the problem setting in this paper.

- W2: Similar to the previous point, the authors do not explain why their **method** should be particularly effective for RAG outputs or outputs with a provided context as opposed to generic LLM outputs. The method does not seem to make use of the provided context in the input prompt for more principled hallucination detection. Instead, the method is similar to existing hallucination detectors from SAE features, albeit with a slightly improved feature selection approach and linear model implementation for prediction.

Minor:
- W3: The hallucination “mitigation” approach appears somewhat misrepresented: the entire (potentially hallucinated) output needs to be generated before the “mitigation” can begin. Since the authors have access to the internals of the models, some intervention or steering prior to generation in order to mitigate hallucinations might be more useful.
W4: It appears that the Figure 4 x-axis labels are incorrect for K’...they show 2e7-2e3, contradicting the text which indicates that K’ is compared from 1024 to 1.
- W5: The SAE architecture and objective should be explained in more detail, as it is essential to the proposed method. This would help with readability.

[1] Ferrando, Javier, et al. "Do I Know This Entity? Knowledge Awareness and Hallucinations in Language Models." arXiv preprint arXiv:2411.14257 (2024).

[2] Suresh, Praneet, et al. "From Noise to Narrative: Tracing the Origins of Hallucinations in Transformers." arXiv preprint arXiv:2509.06938 (2025).

[3] Abdaljalil, Samir, et al. "Safe: A sparse autoencoder-based framework for robust query enrichment and hallucination mitigation in llms." arXiv preprint arXiv:2503.03032 (2025).

[4] Tillman, Henk, and Dan Mossing. "Investigating task-specific prompts and sparse autoencoders for activation monitoring." arXiv preprint arXiv:2504.20271 (2025).

[5] Xin, Chunlei, et al. "Sparse latents steer retrieval-augmented generation." Proceedings of the 63rd Annual Meeting of the Association for Computational Linguistics (Volume 1: Long Papers). 2025.

**Questions:**

- Q1: Why should RAG output hallucinations be considered separately from other types of hallucination? The authors should motivate their proposed method over existing SAE-based hallucination detectors.
- Q2: What makes the proposed method particularly suitable for detecting hallucinations in RAG or grounded output generations if it does not incorporate information from the context?
- Q3: Is it possible to mitigate hallucinations in outputs before generation? Can this be done with the proposed method?

---

> ### Author Response · Authors · 2025-11-20
>
> Thank you for the detailed and constructive feedback. Below we respond point-by-point and describe changes made in the revised manuscript.
>
> > W1, Q1: why RAG hallucinations should be treated differently from generic LLM hallucinations
>
> **########## Clarification of RAG-specific Hallucination Problem ##########**
>
> While recent work has applied SAEs to detect generic LLM hallucinations [1–5], hallucinations in RAG settings present distinct challenges due to the interplay between retrieved evidence and generated content. Generic hallucination detection typically focuses on whether the generated output matches external or world knowledge, often requiring access to reference models or external databases to verify factuality.
>
> In contrast, RAG hallucination (unfaithfulness) is defined strictly with respect to the retrieved context. An output is considered unfaithful if it contradicts or is unsupported by the provided context, regardless of its truth in the real world. This makes RAG hallucination detection a self-contained problem, relying solely on the information given to the model rather than external or parametric knowledge, which may be incomplete or outdated.
>
> We have clarified these distinctions and the unique challenges of RAG hallucination in our revised Introduction, contrasting our problem setting with prior SAE-based hallucination detection work and motivating our research focus.
>
> [1] Ferrando, Javier, et al. "Do I Know This Entity? Knowledge Awareness and Hallucinations in Language Models." arXiv preprint arXiv:2411.14257 (2024).
>
> [2] Suresh, Praneet, et al. "From Noise to Narrative: Tracing the Origins of Hallucinations in Transformers." arXiv preprint arXiv:2509.06938 (2025).
>
> [3] Abdaljalil, Samir, et al. "Safe: A sparse autoencoder-based framework for robust query enrichment and hallucination mitigation in llms." arXiv preprint arXiv:2503.03032 (2025).
>
> [4] Tillman, Henk, and Dan Mossing. "Investigating task-specific prompts and sparse autoencoders for activation monitoring." arXiv preprint arXiv:2504.20271 (2025).
>
> [5] Xin, Chunlei, et al. "Sparse latents steer retrieval-augmented generation." Proceedings of the 63rd Annual Meeting of the Association for Computational Linguistics (Volume 1: Long Papers). 2025\.
>
> > W2, Q2: why their method should be particularly effective for RAG outputs [...] not seem to make use of the provided context [...]
>
> **########## Clarification on Method Design and Novelty ##########**
>
> Although our method does not directly encode the hidden states of the retrieved passages \(C\), the output hidden states for \(y\) are always conditioned on \(C\). This means the SAE features inherently capture interactions between the generated answer and the retrieved context. For RAG hallucination detection, analyzing SAE features on these context-conditioned output states is particularly effective: the features encode context-output relationships, which are precisely what is needed to identify unfaithfulness. In contrast, generic hallucination detection using SAEs relies on the backbone LLM's parametric knowledge to spot mismatches between output tokens and internal knowledge, which may be incomplete or outdated. Our revised Methodology section now includes this clarification.
>
> While applying SAEs to hidden states is established, our contribution lies in systematically addressing the underexplored challenge of RAG-specific hallucination detection. Recent SAE research has advanced mechanistic interpretability for LLMs, but it remains unclear whether sparse autoencoder features can reliably signal RAG-specific hallucinations, which involve complex interactions between retrieved evidence and generated content. RAGLens is, to our knowledge, the first approach to comprehensively demonstrate the effectiveness of SAE-derived features for RAG hallucination detection. We also provide a detailed analysis of design choices, offering practical guidance for optimizing SAE-based detectors for both accuracy and interpretability in RAG scenarios. The revised Introduction now highlights the novelty of this problem and our contributions.

---

> ### Author Response · Authors · 2025-11-20
>
> > W3, Q3: "mitigation" approach appears somewhat misrepresented [...] steering prior to generation [...]
>
> Thank you for the suggestion. We first clarify why our approach emphasizes post-hoc mitigation rather than direct steering of SAE features during generation, and then illustrate how RAGLens can support model steering when suitable features are available.
>
> **########## Rationale for Post-hoc Mitigation ##########**
>
> Our interpretation examples (see Tables 3, 9) show that SAE features effective for detecting RAG hallucinations are often activated concurrently with or after hallucinated tokens are generated. This timing makes direct intervention impractical for prevention, as the undesired content has already appeared. While causal intervention is possible in specific cases, it is not a universal solution for hallucination mitigation. This motivates our focus on post-hoc, text-based feedback in the main mitigation pipeline.
>
> **########## Causal Intervention of SAE Features ##########**
>
> When an SAE feature reliably activates before hallucinated content, it can be used for steering. For instance, Feature 22790 in Llama3.1-8B consistently fires prior to hallucinated numeric or temporal details (e.g., on "of" in the prefix "[...] at the age of"). Suppressing this feature (setting its value to 0 or −20) leads the model to continue with unsupported numbers, while amplifying it (setting to 20) steers the model toward faithful completions, such as using unspecified ages or time frames. This demonstrates that Feature 22790 encodes the model's awareness of potentially hallucinated specifics, and that increasing its activation encourages more faithful output.
>
> | Context | Prefix | Value | Output |
> | :---- | :---- | :---- | :---- |
> | No mention of the age | *[...] at the age of* | −20.0 | *[...] of 30.* |
> |  |  | 0.0 | *[...] of 25.* |
> |  |  | 20.0 | *[...] of an unspecified age.* |
> | No mention of the release date | *[...] is scheduled to be released in* | −20.0 | *[...] in 2016.* |
> |  |  | 0.0 | *[...] in the future.* |
> |  |  | 20.0 | *[...] in an unspecified time frame.* |
>
> We examined another feature from Llama2-13B, which is linked to hallucinations about opening hours and ratings. This feature typically activates only as the hallucinated detail is being produced (e.g., "Monday" in an inconsistent hours statement). By perturbing the feature value on the token before the hallucinated word (e.g., "on" in "on Mondays"), we also confirmed from this example that manipulating SAE features identified by RAGLens can steer model behavior and demonstrate a causal relationship with RAG hallucinations.
>
> | Context | Prefix | Value | Output |
> | :---- | :---- | :---- | :---- |
> | *[...] "Monday": "0:0-0:0"; "Tuesday": "11:0-16:0" "Wednesday": "11:0-21:0" [...]* | *[...] Thai, Restaurants\n\nHours:* | −20.0 | *[...] : the restaurant is open from 11:00 AM to 21:00 PM, Monday to Sunday* |
> |  |  | 0.0 | *[...] : the restaurant is open from 11:00 AM to 21:00 PM, Monday to Sunday* |
> |  |  | 20.0 | *[...] : Monday: 0:00 − 0:00 [...]* |
> | *[...] "Monday": "0:0-0:0" [...]* | *[...] no information available for the business's hours on* | −20.0 | *[...] on hours of operation on Mondays.* |
> |  |  | 0.0 | *[...] on hours of operation on Mondays.* |
> |  |  | 20.0 | *[...] on holidays.* |
>
> These results and further discussion are included in Appendix C of our revised paper.
>
> > W4: Figure 4 x-axis labels are incorrect
>
> Thank you for catching this error. We have updated Figure 4 so that the x-axis now correctly displays log₂(K'), with tick labels ranging from 10 to 0, corresponding to K' values from 1024 down to 1.
>
> > W5: The SAE architecture and objective should be explained in more detail
>
> Thank you for the suggestion. We have now included the detailed SAE architecture information of used SAEs in the Implementation Details section (Appendix B).

---

> ### Comment · Reviewer_sTa7 · 2025-11-24
>
> I thank the authors for their comprehensive responses to my concerns. While I appreciate the added causal intervention examples, I still remain unconvinced as to the uniqueness of the problem setting and the specificity of the proposed method for this problem setting.
>
> **Clarification of RAG-specific Hallucination Problem**
>
> > RAG hallucination (unfaithfulness) is defined strictly with respect to the retrieved context
>
> While this may differentiate the RAG setting from factual hallucinations or errors, maintaining faithfulness to the context is in no way unique to RAG. For instance, a simple summarization task would necessitate the exact same faithfulness with respect to the context. This summarization faithfulness has been explored with SAEs previously. Maintaining consistency in a generated story or long snippet of code would also fall under this category. The RAG setting does not seem unique to me, and since the proposed method does not take into account HOW the context was retrieved, I struggle to see how this is a novel contribution.
>
> **Clarification on Method Design and Novelty**
>
> > Although our method does not directly encode the hidden states of the retrieved passages (C), the output hidden states for (y) are always conditioned on (C). This means the SAE features inherently capture interactions between the generated answer and the retrieved context. For RAG hallucination detection, analyzing SAE features on these context-conditioned output states is particularly effective: the features encode context-output relationships, which are precisely what is needed to identify unfaithfulness. In contrast, generic hallucination detection using SAEs relies on the backbone LLM's parametric knowledge to spot mismatches between output tokens and internal knowledge, which may be incomplete or outdated.
>
> Again, this does not explain why *your method in particular* is well-suited for identifying RAG hallucinations. Why should the proposed method be better than other SAE-based methods for analyzing hallucinations? They would all use the same activations (y), which as you point out are always conditioned on the context (C) regardless of method. While it is true that other papers do not focus on RAG outputs specifically, it is unclear why RAG presents a new paradigm for hallucinations compared to something like summarization, which has been explored.

---

> ### Author Response · Authors · 2025-11-25
> **Follow-up Response (1/2)**
>
> Thank you again for your continued engagement and thoughtful feedback. We are glad that the additional results have addressed some of your concerns. Below, we respond to your remaining points and describe how we have revised the paper to more clearly position our contribution within the existing literature.
>
> ### Clarification of RAG-specific Hallucination Problem
>
> > [...] maintaining faithfulness to the context is in no way unique to RAG [...] a simple summarization task would necessitate the exact same faithfulness [...]
>
> > This summarization faithfulness has been explored with SAEs previously.
>
> Yes, we agree that faithfulness to the provided context is a general requirement for any context-conditioned generation task. In our paper, we use the term "RAG faithfulness" to refer exactly to the faithfulness of LLM outputs to the given context, encompassing not only question answering but also tasks such as summarization and data-to-text generation (e.g., RAGTruth, AggreFact, TofuEval in our Experiments). Our use of "RAG hallucination" follows the conventions of recent literature [1,2,3], where "RAG" broadly denotes any scenario in which generation is conditioned on a supplied context, regardless of whether the retrieval step is nontrivial or the context is fixed. For example, RAGTruth [1] treats summarization as a RAG task by providing a fixed instruction and randomly selected documents as context.
>
> While prior work has explored faithfulness in summarization [4,5,6], to our knowledge, ***the use of SAEs for detecting or analyzing summarization faithfulness remains largely unexplored***. Our work contributes **an early exploration** of SAE-based methods for context-aware faithfulness detection and demonstrates **superior effectiveness** compared to prompting-based, uncertainty-based, or internal representation-based detectors, as evidenced by the performance comparison on RAGTruth in Table 1 of our paper.
>
> In the revised manuscript, we clarify that "RAG" in our work refers generally to LLM generation conditioned on a provided context (see Section 3.1, Problem Setting). Also, our experiments span beyond traditional QA to include summarization and data-to-text generation, covering multiple types of context-conditioned generation tasks (see Appendix A.1).
>
>
> [1] RAGTruth: A Hallucination Corpus for Developing Trustworthy Retrieval-Augmented Language Models. ACL. 2024.
>
> [2] ReDeEP: Detecting Hallucination in Retrieval-Augmented Generation via Mechanistic Interpretability. ICLR. 2025.
>
> [3] RAG-HAT: A hallucination-aware tuning pipeline for LLM in retrieval-augmented generation. EMNLP. 2024.
>
> [4] Evaluating the factual consistency of abstractive text summarization. EMNLP. 2020.
>
> [5] Understanding factual errors in summarization: Errors, summarizers, datasets, error detectors. ACL. 2023.
>
> [6] Tofueval: Evaluating hallucinations of llms on topic-focused dialogue summarization. NAACL. 2024.
>
> > [...] the proposed method does not take into account HOW the context was retrieved [...]
>
> While retrieval is essential for conditioning text generation in RAG, **the detection of RAG faithfulness depends solely on the alignment between the provided context and the generated output**. For this reason, our method is intentionally retrieval-agnostic: RAGLens operates on the context and the generated answer, and can be applied both to classical RAG pipelines and to settings where the context is fixed (e.g., summarization, data-to-text).

---

> ### Author Response · Authors · 2025-11-25
> **Follow-up Response (2/2)**
>
> ### Clarification on Method Design and Novelty
>
> > Why should the proposed method be better than other SAE-based methods for analyzing hallucinations?
>
> Thank you for this insightful question. While our innovation lies in the new exploration of SAEs for RAG faithfulness detection rather than inventing an entirely new SAE-based technique, it is indeed valuable to clarify whether our pipeline (max pooling with information-based feature selection) offers advantages over other SAE-based hallucination detection methods, such as using SAE features from the last token [1,2,3] or max-pooled features without feature selection [4].
>
> To address this, we conducted experiments on Llama2-7B and compared our approach (max-pooled SAE features + feature selection) against two alternatives: (1) using SAE features from the last token, and (2) using all max-pooled features without feature selection. For the latter, we trained a logistic regression (LR) classifier due to the high training cost on all features, and compared it to a simplified RAGLens variant (Max Pooled + Feature Selection + LR).
>
> The results below show that max pooling combined with information-based feature selection consistently outperforms both the last-token baseline and the "all features" logistic regression baseline for hallucination detection. Notably, selecting the top features preserves nearly all relevant information for accurate detection, demonstrating both the effectiveness and efficiency of our design pipeline for RAG hallucination detection. We have added this discussion to the Appendix K (Table 13) of our revised paper, clarifying the advantage of our current design over other SAE-based methods.
>
> | Feature Source | Selection | Classifier | RAGTruth-Summary (Acc) | RAGTruth-Summary (F1) | RAGTruth-Summary (AUC) | RAGTruth-QA (Acc) | RAGTruth-QA (F1) | RAGTruth-QA (AUC) | RAGTruth-Data2txt (Acc) | RAGTruth-Data2txt (F1) | RAGTruth-Data2txt (AUC) |
> |----------------|-----------|------------|------------------------|-----------------------|------------------------|-------------------|------------------|-------------------|-------------------------|------------------------|-------------------------|
> | Last Token     | Yes       | GAM        | 0.6293                 | 0.6462                | 0.7507                 | 0.6908            | 0.6899           | 0.8101            | 0.7454                  | 0.7554                 | 0.8296                  |
> | Max Pooled     | No        | LR         | 0.6734                 | 0.6773                | 0.7305                 | 0.7356            | 0.7152           | 0.8344            | 0.7499                  | 0.7549                 | 0.8397                  |
> | Max Pooled     | Yes       | LR         | 0.6718                 | 0.6724                | 0.7663                 | 0.7572            | 0.7278           | 0.8472            | 0.7085                  | 0.7171                 | 0.8014                  |
> | Max Pooled     | Yes       | GAM        | 0.6973                 | 0.7197                | 0.8191                 | 0.7717            | 0.7678           | 0.8835            | 0.7668                  | 0.7746                 | 0.8454                  |
>
> [1] Do I Know This Entity? Knowledge Awareness and Hallucinations in Language Models. ICLR. 2025.
>
> [2] Investigating task-specific prompts and sparse autoencoders for activation monitoring. Arxiv. 2025.
>
> [3] Sparse latents steer retrieval-augmented generation. ACL. 2025.
>
> [4] From Noise to Narrative: Tracing the Origins of Hallucinations in Transformers. NeurIPS. 2025.
>
> > [...] it is unclear why RAG presents a new paradigm for hallucinations compared to something like summarization [...]
>
> As discussed in our earlier response, our use of the term "RAG hallucination" aligns with recent literature [1,2,3], where "RAG" broadly encompasses any context-conditioned LLM generation, including tasks such as summarization and data-to-text. We have clarified this terminology in the revised paper (Section 3.1) to avoid ambiguity.
>
> [1] RAGTruth: A Hallucination Corpus for Developing Trustworthy Retrieval-Augmented Language Models. ACL. 2024.
>
> [2] ReDeEP: Detecting Hallucination in Retrieval-Augmented Generation via Mechanistic Interpretability. ICLR. 2025.
>
> [3] RAG-HAT: A hallucination-aware tuning pipeline for LLM in retrieval-augmented generation. EMNLP. 2024.

---

### Official Review · Reviewer_uJwo · 2025-10-29

**Soundness:** 3
**Presentation:** 3
**Contribution:** 2
**Rating:** 4
**Confidence:** 4

**Summary:**

This paper addresses the problem of hallucination in RAG systems, where large language models may produce content that contradicts retrieved evidence. The authors propose **RAGLens**, a lightweight and interpretable hallucination detector that leverages sparse autoencoders (SAEs) to extract semantically meaningful internal features predictive of unfaithfulness. Through information-based feature selection and additive feature modeling, RAGLens achieves superior detection accuracy compared to existing methods on the same LLM.

The contributions are: 1. It demonstrates that SAEs can capture nuanced RAG-specific hallucination features; 2. It introduces an interpretable and efficient detector that enables both accurate detection and explainable mitigation; 3. It provides empirical insights into where and how hallucination-related signals emerge within LLMs.

**Strengths:**

Unlike previous approaches, this paper takes an innovative direction by incorporating SAE-derived features for hallucination detection. This approach not only enables accurate identification of hallucinations but also offers interpretable explanations for their generation, making the proposed framework both elegant and effective.

Empirical results show that RAGLens achieves superior performance on two hallucination benchmarks compared with multiple baselines. The ablation studies further validate the soundness of key design choices, such as using a GAM as the predictor.

Moreover, the theoretical analysis, including the proof that max pooling can distinguish hallucination-related activations, provides additional confidence in the reliability of the proposed method.

**Weaknesses:**

A key concern is that the paper does not clearly establish the causal connection between SAE features and hallucination behavior. Although the theoretical analysis argues that, under sparse activation, max pooling can amplify hallucination-related signals and suppress noise, it remains unclear whether the improved detection performance primarily stems from the SAE-derived features or from the predictive capacity of the GAM. If the authors could provide stronger evidence that the learned SAE features themselves are systematically associated with hallucination modes, the technical contribution would be more convincing.

In addition, while the main experiments include a reasonable set of baselines, the evaluation is limited to two LLaMA-based models (LLaMA2-7B and LLaMA2-13B). Although Section 4.3 reports results on other models, the comparisons are restricted to the CoT-based evaluation setting. This narrow range of model types raises concerns about the generalizability of the proposed method across a wide range of LLMs.

**Questions:**

1. In the $\textbf{Analysis of Feature Count}$ section, why does the number of $K^{'}$ have such different effects across datasets? It has a significant impact on TofuEval, but its effect on AggreFact is much less pronounced.

2. I would appreciate it if you could provide a more detailed explanation of how $\textbf{Global Explanations}$ are realized, as I find this part unclear.

---

> ### Author Response · Authors · 2025-11-20
>
> Thank you for the detailed and constructive feedback. Below we respond point-by-point and describe changes made in the revised manuscript.
>
> > causal connection between SAE features and hallucination behavior [...] whether the improved detection performance primarily stems from the SAE-derived features or from the predictive capacity of the GAM [...] evidence that the learned SAE features themselves are systematically associated with hallucination modes [...]
>
> **########## Analysis of SAE Feature vs. Hidden State ##########**
>
> Thank you for the insightful question. As demonstrated in the predictor ablation study (Figure 5), the GAM classifier plays a significant role in RAGLens's strong performance. To further disentangle the contributions of SAE-derived features, we conducted an experiment where we replaced SAE features with the raw hidden states from Llama2-7B, while keeping the MI-based feature selection and GAM classifier unchanged. The table below compares classification accuracy across varying feature dimensions ($K'$) for both hidden states and SAE features.
>
> | Data | Source | K'=1024 | K'=512 | K'=256 | K'=128 | K'=64 | K'=32 | K'=16 | K'=8 | K'=4 | K'=2 | K'=1 |
> | :---- | :---- | :---- | :---- | :---- | :---- | :---- | :---- | :---- | :---- | :---- | :---- | :---- |
> | RAGTruth | Hidden | 0.7851 | 0.7624 | 0.7639 | 0.7570 | 0.7462 | 0.7462 | 0.7417 | 0.7334 | 0.7180 | 0.7071 | 0.6750 |
> |  | SAE | 0.7998 | 0.7922 | 0.7893 | 0.7829 | 0.7637 | 0.7611 | 0.7549 | 0.7467 | 0.7400 | 0.7289  | 0.7072 |
> | AggreFact | Hidden | 0.7150 | 0.7040 | 0.7035 | 0.6725 | 0.6801 | 0.6668 | 0.6751 | 0.6584 | 0.6687  | 0.6283 | 0.5584 |
> |  | SAE | 0.7039 | 0.6909  | 0.6907 | 0.6912 | 0.6976 | 0.6945 | 0.6891 | 0.6814 | 0.6905 | 0.6763 | 0.6774 |
> | TofuEval | Hidden | 0.6991 | 0.6868 | 0.6596 | 0.6308 | 0.6379 | 0.6374 | 0.6229 | 0.5979 | 0.6089 | 0.5847 | 0.5000 |
> |  | SAE | 0.6932 | 0.6846 | 0.7063 | 0.6736 | 0.6935 | 0.6973 | 0.7007 | 0.6908 | 0.6911 | 0.6452 | 0.5000 |
>
> We observe that with a large $K'$, hidden states perform similarly to SAE features, which is expected since SAE features are derived from hidden states. However, as $K'$ decreases, the accuracy of hidden states drops much more sharply, especially on single-task datasets like AggreFact and TofuEval. This suggests that SAE features are more effective at isolating hallucination-related signals and concentrating them into a compact set of dimensions. Such compactness not only enhances interpretability but also facilitates downstream tasks like hallucination mitigation, as monitoring a small number of salient features is sufficient for effective control. We have added a visualized version of this comparison and extended results on additional metrics in Appendix D.
>
> Moreover, our qualitative analyses in Appendix H show that the identified SAE features correspond to coherent RAG-specific hallucination modes (e.g., "unsupported numeric/time specifics"). Their activations change predictably under context perturbations that add/remove supporting evidence. These interventions confirm that these features are closely and systematically linked to hallucination behavior, rather than being arbitrarily correlated signals.

---

> ### Author Response · Authors · 2025-11-20
>
> > the evaluation is limited to two LLaMA-based models
>
> **########## Baseline Comparison on More Backbone LLMs ##########**
>
> Our primary comparison with ReDeEP and other internal-state-based baselines focuses on Llama2-7B and Llama2-13B, since these methods require access to internal activations from the same backbone LLM used to generate the benchmark dataset examples. RAG hallucination annotations are available for Llama2-7B/13B in RAGTruth and Dolly, but not for the additional models evaluated in Section 4.3. Consequently, experiments with those models use the CoT-based evaluation setting rather than direct comparison with internal-state–based baselines.
>
> To further evaluate the generalizability of RAGLens beyond Llama2, we conducted an additional experiment using Llama3-8B. Leveraging third-party annotations of Llama3-8B generations on RAGTruth and Dolly released by ReDeEP, we assessed AUC for RAGLens and compared it with other baselines. As shown below, RAGLens consistently outperforms all baselines on both datasets, demonstrating strong performance and robust applicability to newer LLMs:
>
> | Method | RAGTruth | Dolly |
> | :---- | :---- | :---- |
> | RAGAS | 0.6776 | 0.6870 |
> | SEP | 0.7004 | 0.5159 |
> | SAPLMA | 0.7092 | 0.5019 |
> | ITI  | 0.6534 | 0.5011 |
> | ReDeEP | 0.7285 | 0.7354 |
> | RAGLens (Ours) | 0.7738 | 0.8284 |
>
> > why does the number of K' have such different effects across datasets
>
> **########## Analysis of Data-specific Effects of K' ##########**
>
> As shown in Figure 4, the impact of varying K' differs across datasets: RAGTruth exhibits a gradual decline in performance as K' decreases, while AggreFact and TofuEval maintain stable accuracy until K' becomes very small. This discrepancy arises from the datasets' task diversity. RAGTruth contains multiple subtasks (summarization, QA, data-to-text), requiring a broader set of SAE features to capture hallucination signals. Reducing K' in RAGTruth progressively removes features important for specific subtasks, resulting in a smoother performance drop. In contrast, AggreFact and TofuEval focus on single, homogeneous tasks, so fewer features are needed and performance remains robust as K' decreases, only dropping sharply when the feature set becomes insufficient.
>
> > detailed explanation of how Global Explanations are realized
>
> **########## Implementation of Global Explanations ##########**
>
> Our global explanations build on two components: (i) the dictionary-learning property of SAEs, and (ii) the additive structure of the GAM classifier. We first collect representative activation cases for each selected feature by sampling 12 top-activated examples and 12 quantile-distributed examples based on the actual activations in the RAGTruth training data. These activation cases are then used to summarize the semantic concept encoded by each feature using GPT-5, using the prompt template shown in Figure 7.
>
> In parallel, for each selected feature, we visualize the one-dimensional shape function learned by the GAM, which maps the feature's activation magnitude to its contribution to the predicted hallucination risk. Together, the semantic summary (from representative activations) and the corresponding shape function (from the GAM) provide an instance-invariant, feature-level explanation. They describe what concept the feature captures and how increases or decreases in that concept globally affect the hallucination prediction across examples.
>
> We have clarified and detailed this procedure in the revised Implementation Details section (Appendix B).

---

### Official Review · Reviewer_AJ7o · 2025-10-30

**Soundness:** 3
**Presentation:** 3
**Contribution:** 3
**Rating:** 6
**Confidence:** 3

**Summary:**

This paper proposes a framework leveraging Sparse Autoencoders (SAEs) to analyze and mitigate hallucinations in Retrieval-Augmented Generation (RAG) systems. The authors argue that hallucinations arise from misaligned interactions between parametric knowledge and external context, which can be disentangled through feature-level representation learning. By training SAEs on internal activations of RAG models, the paper identifies sparse and interpretable features correlated with faithful or hallucinatory outputs. The authors further construct a faithfulness detection model based on these features and explore case studies of interpretable neurons linked to reasoning or retrieval grounding. Experiments on RAGTruth and HalluRAG show improvements in detecting hallucinations and understanding model behaviors, demonstrating the potential of feature-level interpretability for trustworthy RAG.

**Strengths:**

1.	Innovative use of sparse autoencoders to dissect RAG internal activations and link specific features to hallucination phenomena.
2.	Interpretability focus — provides human-understandable explanations and visualizations for neuron activations contributing to hallucination or faithful grounding.
3.	Comprehensive empirical setup, covering both quantitative metrics and qualitative feature studies on multiple benchmarks.
4.	Clear motivation and relevance to improving trustworthy and explainable RAG systems.

**Weaknesses:**

1.	Lack of causal intervention — The paper provides interpretive insights but does not test whether manipulating identified SAE features can reduce hallucinations. Given that SAE features allow for activation-level control, it would be valuable to explore feature interventions to demonstrate causality.
2.	Unclear explanation data source — In Section 4.4, the authors show two representative features and mention that the semantic explanations were distilled from 24 activation cases. However, it is not clear where these cases originate. If they are from the RAGTruth dataset, they may not generalize beyond it; if they are from the pretraining corpus, examples should be shown in the paper to ensure explanation reliability.
3.	Potential bias in interpretation — Since the 24 examples are used to generate explanations, the feature semantics may reflect dataset-level biases. The authors could validate their explanations quantitatively, e.g., by correlating simulated activations (based on explanation-derived templates) with real activations to show consistency.
4.	Ambiguity in faithfulness detection — The method relies on features such as ID 22790, which are activated when models refer to external context. Yet, if the model’s internal and external knowledge are aligned, the same feature might be activated even without external grounding, potentially causing false positives. Similarly, when external information is incomplete or noisy, the model may rely partly on internal reasoning—raising questions about whether the proposed detection scheme remains reliable.
5.	Limited generalization discussion — The current framework is primarily designed for RAG tasks with correct retrievals. The paper should further discuss or test its applicability under retrieval noise, domain shifts, or reasoning-heavy tasks.

**Questions:**

1.	Can the authors test whether directly intervening on selected SAE features (e.g., amplifying or suppressing feature 22790) reduces hallucinations, to demonstrate causal interpretability?
2.	Where do the 24 activation cases used for GPT-5 explanation distillation come from? Are they drawn from RAGTruth, or from broader pretraining corpora?
3.	Have the authors examined whether high activations in the pretraining corpus correspond to similar semantics as in RAGTruth? If so, do the feature functions remain consistent?
4.	Could the authors quantify explanation reliability—for example, by testing correlation between explanation-based simulated activations and real activations across large samples?
5.	How does the feature-based faithfulness detector perform when external knowledge is incomplete or partially conflicting with internal knowledge?

---

> ### Author Response · Authors · 2025-11-20
>
> Thank you for the detailed and constructive feedback. Below we respond point-by-point and describe changes made in the revised manuscript.
>
> > W1, Q1: Lack of causal intervention
>
> **########## Causal Intervention of SAE Features ##########**
>
> Building on this suggestion, we conducted controlled interventions on Feature 22790 in Llama3.1-8B to examine its causal impact on model outputs. Feature 22790 is reliably activated just before the generation of hallucinated numeric or temporal details, such as on the token "of" in the answer prefix "... at the age of", which precedes unsupported numeric content. To test causality, we held the prompt, context, and answer prefix fixed, intervened only on the value of Feature 22790 at its activation position, and left all other activations unchanged. By varying the feature's value (e.g., −20, 0, 20), we observed the model's subsequent generations.
>
> As the table below shows, suppressing the feature (setting its value to 0 or −20) led the model to continue with hallucinated, unsupported numbers. In contrast, amplifying the feature (setting it to 20) steered the model toward faithful completions, such as using unspecified ages or time frames instead of hallucinated specifics. These results indicate that Feature 22790 encodes the model's awareness of potentially hallucinated numeric or temporal content, and that increasing its activation can promote more faithful output in these scenarios.
>
> | Context | Prefix | Value | Output |
> | :---- | :---- | :---- | :---- |
> | No mention of the age | *[...] at the age of* | \-20.0 | *[...] of 30\.* |
> |  |  | 0.0 | *[...] of 25\.* |
> |  |  | 20.0 | *[...] of an unspecified age.* |
> | No mention of the release date | *[...] is scheduled to be released in* | \-20.0 | *[...] in 2016\.* |
> |  |  | 0.0 | *[...] in the future.* |
> |  |  | 20.0 | *[...] in an unspecified time frame.* |
>
> While our analysis of Feature 22790 demonstrates its causal role in hallucination mitigation, interpretation examples in our paper (e.g., Tables 3, 9) reveal that SAE features useful for detecting RAG hallucinations are not always activated prior to the generation of hallucinated text. In many cases, these features only fire concurrently with or after the hallucinated tokens have already been produced. As a result, direct intervention on such features is impractical for preventing hallucinations, since the undesired content has already appeared by the time these activations occur. Therefore, although causal intervention on SAE features is possible in specific scenarios, it is not a universal solution for hallucination mitigation. This limitation motivates our emphasis on post-hoc, text-based feedback in the main mitigation pipeline. We have included the intervention results and the discussion in Appendix C of our revised paper.
>
> > W2, Q2, Q3: Unclear explanation data source [...] activations in the pretraining corpus
>
> **########## Implementation and Verification of SAE Interpretations ##########**
>
> To summarize the semantic meaning of each SAE feature, we selected 24 representative cases from the RAGTruth training data: 12 instances with the highest activation values for the feature and 12 additional instances distributed across the activation quantiles. This approach ensures coverage of both strong and moderate activations, providing a broad perspective on the feature's behavior. GPT-5 is then prompted with the template in Figure 7 to summarize the feature meaning. We have now clarified this procedure in Appendix B (Implementation Details).
>
> To assess whether the semantics of Feature 22790 generalize beyond RAGTruth, we computed its activations on the first 10,000 examples from the lmsys/lmsys-chat-1m dataset and inspected the top-activated cases, which are shown below:
>
> | Sample Index | Input | Output |
> | :---- | :---- | :---- |
> | 9384 | TRANSCRIPT='Hello doctor I have fever and cough. Okay take paracetamol and go home and rest.' [...] | [...] Patient advises they have been experiencing symptoms **for the past** two days [...] |
> | 3298 | User: Telekom Deutschland GmbH [...] | [...] "summary": "Mobilfunk-Rechnung für den **Monat** März 2023" |
>
> In both RAGTruth and lmsys-chat-1m, Feature 22790 consistently activates on tokens introducing numeric or temporal details (e.g., "for the past", "Monat" with month/year), spanning diverse formats such as clinical dialogue transcripts and multiple languages including German. This cross-dataset and cross-lingual consistency supports the interpretation that Feature 22790 robustly encodes the concept of "unsupported numeric/time specifics," rather than reflecting dataset-specific semantics. We have added this analysis to Appendix E.

---

> ### Author Response · Authors · 2025-11-20
>
> > W3, Q4: Potential bias in interpretation
>
> **########## Correlation Analysis of SAE Interpretations ##########**
>
> Following your suggestion, we prompted GPT-5 to predict the activation level of Feature 22790 on 24 held-out RAGTruth test cases (8 top-activated and 16 quantile-distributed examples), using the distilled natural-language explanation. For each case, the three top-activated tokens were highlighted, and GPT-5 was asked to assign a score from 0 (feature not present) to 5 (very strong match). We then compared these predicted scores to the actual SAE activations. The resulting Pearson correlation was 0.6731 (p < 0.05), indicating that the explanation provides a reasonably accurate description of when the feature should fire across diverse examples. This reduces concern that the explanation is merely fitting a few cherry-picked cases.
>
> We also acknowledge that, while SAEs are designed to disentangle distinct semantic concepts from hidden states, some SAE features may remain generic or polysemantic, limiting their interpretability, which is a challenge widely recognized in current SAE research \[1,2\]. We have added this new analysis and discussion to Appendix E, noting that the architecture-agnostic design of RAGLens allows it to benefit from future advances in SAE methods, which may enable even more transparent and effective detectors for RAG hallucinations.
>
> \[1\] Towards Monosemanticity: Decomposing Language Models with Dictionary Learning. 2023\.
>
> \[2\] Sparse Autoencoders Find Highly Interpretable Features in Language Models. 2024\.
>
> > W4, Q5: Ambiguity in faithfulness detection [...] external information is incomplete or noisy [...]
>
> **########## Clarifications on Faithfulness Detection ##########**
>
> We would like to address this concern in two ways. First, RAGLens relies on multiple SAE features rather than a single feature, which reduces sensitivity to false positives from any individual feature. Second, we conducted counterfactual analyses to directly test for false positive activations when the context is removed or modified (see Appendix H, Tables 10–12). Specifically, we recomputed feature activations on the same answer after removing the retrieval context. As shown in Table 11 of the revised paper, the relevant SAE feature activations collapse when the context is absent, indicating that these features are RAG-specific and not generic detectors of numeric tokens. To further evaluate robustness under noisy evidence, we perturbed the context by introducing partially conflicting passages. Tables 10–12 demonstrate that hallucination-related features are strongly activated only when the generated answer contradicts or extrapolates beyond the perturbed context. These results support that RAGLens detects mismatches between output and context, rather than simply responding to the generated tokens.

---

> ### Author Response · Authors · 2025-11-20
>
> > W5, Q5: Limited generalization discussion [...] applicability under retrieval noise, domain shifts, or reasoning-heavy tasks.
>
> Thank you for the insightful suggestions. As discussed in our response to W4 and Q5, our counterfactual and perturbation experiments have already examined RAGLens behavior under partial or corrupted evidence. We agree, however, that a systematic evaluation under controlled retrieval noise and multi-step reasoning is important. Currently, there are few RAG hallucination benchmarks with instance-level labels that explicitly vary retrieval quality or require complex reasoning over retrieved evidence. Addressing these scenarios remains an important direction for future work. In response to your suggestion regarding domain shifts, we have conducted new experiments to further evaluate RAGLens under these conditions.
>
> **########## Analysis of Cross-domain Generalization ##########**
>
> To evaluate how RAGLens generalizes across domain shifts, we added two sets of experiments to directly evaluate generalization across datasets and tasks. Since these are out-of-distribution scenarios, we report AUROC, which is robust to label imbalance and measures whether RAGLens scores remain consistent with ground-truth labels under domain shift.
>
> First, we evaluate cross-dataset generalization by training the GAM classifier on one dataset and testing its performance on other datasets without retraining. The results demonstrate that RAGLens generalizes best when trained on diverse data. For instance, a detector trained on RAGTruth achieves substantially higher AUROC scores than the CoT baseline when applied to AggreFact and TofuEval. In contrast, predictors trained on AggreFact or TofuEval, while often outperforming CoT, show reduced generalization compared to those trained on RAGTruth. This difference is likely due to the broader task coverage in RAGTruth, which includes multiple subtasks, whereas AggreFact and TofuEval are limited to single tasks. Overall, these findings suggest that training RAGLens on diverse datasets enhances its robustness and ability to handle domain shifts.
>
> | Model | Setting | RAGTruth | AggreFact | TofuEval |
> | :---- | :---- | :---- | :---- | :---- |
> | Llama2-7B | CoT | 0.4842 | 0.5741 | 0.5562 |
> |  | Trained on RAGTruth | 0.8806 | 0.8019 | 0.7637 |
> |  | Trained on AggreFact | 0.5330 | 0.8330 | 0.6123 |
> |  | Trained on TofuEval | 0.7747 | 0.6161 | 0.7846 |
> | Llama2-13B | CoT | 0.4959 | 0.5285 | 0.5583 |
> |  | Trained on RAGTruth | 0.8674 | 0.7831 | 0.7319 |
> |  | Trained on AggreFact | 0.4669 | 0.8285 | 0.6239 |
> |  | Trained on TofuEval | 0.7342 | 0.5727 | 0.7883 |
>
> Second, we evaluate cross-task generalization within RAGTruth by training RAGLens on one subtask and testing its performance on other subtasks. The results below show that RAGLens consistently transfers knowledge across tasks and outperforms the CoT baseline. Notably, the predictor trained on summarization (Summary) demonstrates the strongest generalizability, surpassing those trained on question answering (QA) or data-to-text generation (Data2txt). Furthermore, knowledge transfer between Summary and QA is more effective than between Data2txt and the other tasks. These findings indicate that RAGLens can capture signals common to multiple RAG tasks, while also highlighting the existence of task-specific features that constrain generalization.
>
> | Model | Setting | RAGTruth-Summary | RAGTruth-QA | RAGTruth-Data2txt |
> | :---- | :---- | :---- | :---- | :---- |
> | Llama2-7B | CoT | 0.4924 | 0.4845 | 0.4949 |
> |  | Trained on RAGTruth-Summary | 0.8191 | 0.8253 | 0.6443 |
> |  | Trained on RAGTruth-QA | 0.7081 | 0.8835 | 0.6609 |
> |  | Trained on RAGTruth-Data2txt | 0.5386 | 0.6616 | 0.8454 |
> | Llama2-13B | CoT | 0.5196 | 0.5088 | 0.4765 |
> |  | Trained on RAGTruth-Summary | 0.7539 | 0.8330 | 0.6627 |
> |  | Trained on RAGTruth-QA | 0.6619 | 0.8769 | 0.6669 |
> |  | Trained on RAGTruth-Data2txt | 0.5653 | 0.7373 | 0.8491 |
>
> We have now included these analyses as Section 4.4 in our revised paper.

---

> ### Comment · Reviewer_AJ7o · 2025-11-28
>
> I thank the authors for their comprehensive responses. I would like to keep the original score.

---

### Official Review · Reviewer_rg9H · 2025-10-30

**Soundness:** 3
**Presentation:** 2
**Contribution:** 3
**Rating:** 6
**Confidence:** 4

**Summary:**

The paper proposes a way to detect hallucinations in the context of RAG by training a generalized additive model on SAE-encoded feature vectors; token-level activations are channel-wise max pooled to produce instance level features via selection on mutual information. The authors compare against many baselines (though, slightly narrowly selected- they don't compare against pretty area-standard hallucination/factuality detectors like factscore/factcc/summac/stronger model internal state probe baselines) and show that their method performs better, across datasets like RAGTruth, Dolly, Aggrefact, and tofueval. Nonetheless, the approach is shown to reduce hallucination rates in a small mitigation study via token-level and instance-level feedback, by prompting the model to revise its original output with RAGLens' feedback. Ablations show that middle layers perform best in detecting hallucinations (though this isn't a new finding).

**Strengths:**

Novel approach that provides both token-level hallucination detections through SAE features. Empirical results show that this approach outperforms all tested contemporary baselines. The approach is generally very lightweight if you have a trained SAE for the model. The method represents a contribution towards utilizing SAE features for hallucination detection in RAG settings.

**Weaknesses:**

The subsection "4.3 GENERALIZATION ACROSS LLMS" is not greatly named; the authors here only test if each model's own trained GAM outperforms the baseline of the model itself's chain-of-thought explanation of if it hallucinated or not. The method itself, due to needing to be trained on model-specific SAE features, doesn't translate across LLMs- that is, for every LLM, you will need to train your own model. The authors also don't really test generalization *across domains*, even for the same model- that is, how does a GAM trained on one domain (i.e. summarization) translate to detecting hallucinations in another (i.e. dialog).

The comparison models chosen miss out on some pretty standard area approaches for hallucination detection- factscore, FactCC, etc. Furthermore, novelty is limited- the paper largely combines existing ideas (SAEs, MI-based feature selection + additive modeling) rather than introducing a fundamentally new detection mechanism.

The hallucination mitigation approach is not well evaluated. The evaluation is to see if the LLM can use the RAGLens feedback, when prompted, to mitigate the hallucination it caused. To evaluate whether the revisions are better, the paper mostly relies on LLM judges; only 45 examples are evaluated by humans, though the results are convincing and could use greater robustness.

**Questions:**

N/A

---

> ### Author Response · Authors · 2025-11-20
>
> Thank you for the detailed and constructive feedback. Below we respond point-by-point and describe changes made in the revised manuscript.
>
> > "4.3 GENERALIZATION ACROSS LLMS" is not greatly named [...] The method itself [...] doesn't translate across LLMs
>
> **########## Clarification on Cross-model Application ##########**
>
> We agree that "Generalization across LLMs" does not accurately reflect the content of the subsection. Unlike Section 4.2, which evaluates RAGLens using an LLM's internal signals to detect its own hallucinations, Section 4.3 investigates whether RAGLens trained on one model can detect hallucinations in outputs generated by other LLMs. This addresses a key limitation of prior internal-signal-based methods, which typically require access to the internal activations of the same model that produced the output. While SAE features themselves are not transferable across different LLMs, applying RAGLens directly to text outputs enables cross-model detection of hallucinations. In the revised paper, we have updated the section title to "Cross-model Application" and clarified that this subsection explores whether LLMs encode sufficient internal knowledge to identify hallucinations in outputs from other models.
>
> > [...] don't really test generalization across domains
>
> **########## Analysis of Cross-domain Generalization ##########**
>
> We appreciate this suggestion and have added two sets of experiments to directly evaluate generalization across datasets and tasks. Since these are out-of-distribution scenarios, we report AUROC, which is robust to label imbalance and measures whether RAGLens scores remain consistent with ground-truth labels under domain shift.
>
> First, we evaluate cross-dataset generalization by training the GAM classifier on one dataset and testing its performance on other datasets without retraining. The results demonstrate that RAGLens generalizes best when trained on diverse data. For instance, a detector trained on RAGTruth achieves substantially higher AUROC scores than the CoT baseline when applied to AggreFact and TofuEval. In contrast, predictors trained on AggreFact or TofuEval, while often outperforming CoT, show reduced generalization compared to those trained on RAGTruth. This difference is likely due to the broader task coverage in RAGTruth, which includes multiple subtasks, whereas AggreFact and TofuEval are limited to single tasks. Overall, these findings suggest that training RAGLens on diverse datasets enhances its robustness and ability to handle domain shifts.
>
> | Model | Setting | RAGTruth | AggreFact | TofuEval |
> | :---- | :---- | :---- | :---- | :---- |
> | Llama2-7B | CoT | 0.4842 | 0.5741 | 0.5562 |
> |  | Trained on RAGTruth | 0.8806 | 0.8019 | 0.7637 |
> |  | Trained on AggreFact | 0.5330 | 0.8330 | 0.6123 |
> |  | Trained on TofuEval | 0.7747 | 0.6161 | 0.7846 |
> | Llama2-13B | CoT | 0.4959 | 0.5285 | 0.5583 |
> |  | Trained on RAGTruth | 0.8674 | 0.7831 | 0.7319 |
> |  | Trained on AggreFact | 0.4669 | 0.8285 | 0.6239 |
> |  | Trained on TofuEval | 0.7342 | 0.5727 | 0.7883 |
>
> Second, we evaluate cross-task generalization within RAGTruth by training RAGLens on one subtask and testing its performance on other subtasks. The results below show that RAGLens consistently transfers knowledge across tasks and outperforms the CoT baseline. Notably, the predictor trained on summarization (Summary) demonstrates the strongest generalizability, surpassing those trained on question answering (QA) or data-to-text generation (Data2txt). Furthermore, knowledge transfer between Summary and QA is more effective than between Data2txt and the other tasks. These findings indicate that RAGLens can capture signals common to multiple RAG tasks, while also highlighting the existence of task-specific features that constrain generalization.
>
> | Model | Setting | RAGTruth-Summary | RAGTruth-QA | RAGTruth-Data2txt |
> | :---- | :---- | :---- | :---- | :---- |
> | Llama2-7B | CoT | 0.4924 | 0.4845 | 0.4949 |
> |  | Trained on RAGTruth-Summary | 0.8191 | 0.8253 | 0.6443 |
> |  | Trained on RAGTruth-QA | 0.7081 | 0.8835 | 0.6609 |
> |  | Trained on RAGTruth-Data2txt | 0.5386 | 0.6616 | 0.8454 |
> | Llama2-13B | CoT | 0.5196 | 0.5088 | 0.4765 |
> |  | Trained on RAGTruth-Summary | 0.7539 | 0.8330 | 0.6627 |
> |  | Trained on RAGTruth-QA | 0.6619 | 0.8769 | 0.6669 |
> |  | Trained on RAGTruth-Data2txt | 0.5653 | 0.7373 | 0.8491 |
>
> We have now included these analyses as Section 4.4 in our revised paper.

---

> ### Author Response · Authors · 2025-11-20
>
> > [...] miss out on some pretty standard area approaches for hallucination detection
>
> **########## Addition of More Baselines ##########**
>
> Following your suggestion, we have added FActScore and FactCC as standard hallucination detection baselines. The table below presents a direct comparison, with all methods using the same Llama2 backbone as the detector:
>
> | Model | Method | RAGTruth \- AUC | RAGTruth \- Acc | RAGTruth \- F1 | Dolly \- AUC | Dolly \- Acc | Dolly \- F1 |
> | :---- | :---- | :---- | :---- | :---- | :---- | :---- | :---- |
> | Llama2-7B | FActScore  | 0.5428 | 0.5333 | 0.6719 | 0.4813 | 0.5354 | 0.6849 |
> |  | FactCC | 0.4976 | 0.5022 | 0.4589 | 0.6169 | 0.5758 | 0.5882 |
> |  | RAGLens | 0.8413 | 0.7576 | 0.7636  | 0.8764  | 0.7778  | 0.8070 |
> | Llama2-13B | FActScore  | 0.5294 | 0.4533 | 0.6239 | 0.4389 | 0.4646 | 0.5954 |
> |  | FactCC | 0.4753 | 0.4800 | 0.4121 | 0.6496 | 0.6162 | 0.5250 |
> |  | RAGLens | 0.8964  | 0.8333  | 0.8148  | 0.8568  | 0.7576  | 0.7895 |
>
> On both RAGTruth and Dolly, RAGLens consistently outperforms FActScore and FactCC across AUC, accuracy, and F1 metrics. This strengthens the empirical case that internal-signal-based methods, when equipped with interpretable SAE features and GAMs, are more effective than standard reference-based factuality metrics for detecting hallucinations produced by the same LLM. These results have been incorporated into Table 1 in our revised paper.
>
> > the paper largely combines existing ideas [...]
>
> **########## Clarification on Novelty ##########**
>
> The novelty of our work is not in inventing new concepts, but in identifying an underexplored research question and providing a systematic, comprehensive analysis of its solution. Recent SAE research has advanced mechanistic interpretability for LLMs, but it is still unclear whether RAG-specific hallucinations can be effectively detected using sparse autoencoder features. Unlike generic hallucinations in LLMs, hallucinations in RAG settings present unique challenges due to the complex interplay between retrieved evidence and generated content.
>
> To our knowledge, RAGLens is the first approach to systematically show that SAE-derived features are highly effective for hallucination detection in RAG settings. We also offer an in-depth analysis of key design choices, giving practical guidance on optimizing SAE-based detectors for both accuracy and interpretability in the RAG context. The revised Introduction now emphasizes the novelty of this open problem and clarifies our contributions in addressing it.
>
> > The hallucination mitigation approach is not well evaluated [...] the paper mostly relies on LLM judges
>
> **########## Evaluation of Hallucination Mitigation ##########**
>
> We acknowledge this concern. Evaluating hallucination mitigation at scale is challenging, as large-scale human annotation is costly and no automatic metric is perfect. Following recent work \[1, 2\], we use LLM-as-a-judge and supplement it with small-scale human evaluation for cross-validation.
>
> To further address this issue, we added an additional analysis using RAGLens itself as an independent detector, leveraging its high accuracy on the task. Specifically, a detector based on Llama3.1-8B is applied to outputs before and after mitigation. It predicts that instance-level feedback converts 29 previously hallucinated responses into non-hallucinations, while token-level feedback converts 36, mirroring the trends observed with LLM and human judges. We have updated Section 4.6 with these results, providing a new perspective on the effectiveness of RAGLens in mitigating RAG hallucinations.
>
> \[1\] Benchmarking LLM Faithfulness in RAG with Evolving Leaderboards. 2025\.
>
> \[2\] A Survey of Automatic Hallucination Evaluation on Natural Language Generation. 2025\.

---

### Meta-Review · Area_Chair_uEaC · 2026-01-09

**Summary:**

This paper introduces RAGLens, a hallucination detector that to unfaithful RAG outputs using LLM internal representations, using sparse autoencoders.

Reviewers were mostly concerned with the limited contribution of the work and its limited generalizability. SAEs for hallucination detection have already been explored, and MI-based feature selection, GAMs for prediction are existing techniques. While their combination is sensible it is difficult to see how it is innovative. However, it has been systematically applied to this specific problem of RAG hallucinations, with comprehensive evaluation.

Given these assessments, I think this paper could be a borderline accept / reject, with arguments on both sides.

**Reviewer Concerns:**

**Addressed**
1. Rg9h: Cross-domain generalization (comprehensive experiments were provided), missing standard baselines (added facctscore).
2. AJ7o: The authors provided empirical evidence, additional experiments, or an acknowledgment of limitations.
3. Ujwo: Causal connection between SAE features and hallucination, limited model evaluation (added another model), dataset-specific effects, explanation of global explanations
4. Sta7: minor issues and post-hoc vs prevention question.


**Unaddressed**
1. Rg9h, aj70, sta7: Limited novelty / generalization as the paper combines existing ideas (SAEs, MI-based feature selection, GAMs) rather than introducing fundamentally new mechanisms. Authors reframed this as identifying an underexplored research question rather than claiming technical novelty
2. Rg9h: Model-specific training requirement for GAMs. Authors clarified the distinction but didn't resolve the limitation
3. Aj70: intervention timing limitations, lack of suitable benchmarks.
4. Sta7: method not designed specifically for RAG. Authors don't explain why their method is particularly effective for RAG outputs; method doesn't make use of provided context. This is better addressed with empirical evidence in the follow up response, but reviewer's conceptual concern about novelty vs. summarization remains.

**Reviewer Scores:**

- Rg9h: never responded directly to the author response. However, based on the author response, I would determine that this reviewer might maintain or improve their score.
- Aj7o: based on the author response, they decide to keep their original score.
- uJwo: never directly responded to the author response. The authors provided either new experiments (SAE vs hidden states, Llama3-8B evaluation), or clear explanations (K' effects, global explanations) for every concern raised.  Based on the author response, I would determine that this reviewer might improve their score.
-sta7: Despite reviewer / author discussion, it remains unclear if the reviewer can be convinced that the method is making a novel problem formulation and methodological innovation. The reviewer's persistent concerns suggest they value novelty, thus might not improve their score.

---

### Decision · Program_Chairs · 2026-01-26

Accept (Poster)